# QATCH: Benchmarking SQL-centric tasks with Table Representation Learning Models on Your Data

**Simone Papicchio**
Politecnico di Torino
Turin, Italy

**Paolo Papotti**
EURECOM
Sophia Antipolis, France

**Luca Cagliero**
Politecnico di Torino
Turin, Italy

## Abstract

Table Representation Learning (TRL) models are commonly pre-trained on large open-domain datasets comprising millions of tables and then used to address downstream tasks. Choosing the right TRL model to use on proprietary data can be challenging, as the best results depend on the content domain, schema, and data quality. Our purpose is to support end-users in testing TRL models on proprietary data in two established SQL-centric tasks, i.e., Question Answering (QA) and Semantic Parsing (SP). We present QATCH (Query-Aided TRL Checklist), a toolbox to highlight TRL models' strengths and weaknesses on relational tables unseen at training time. For an input table, QATCH automatically generates a testing checklist tailored to QA and SP. Checklist generation is driven by a SQL query engine that crafts tests of different complexity. This design facilitates inherent portability, allowing the checks to be used by alternative models. We also introduce a set of cross-task performance metrics evaluating the TRL model's performance over its output. Finally, we show how QATCH automatically generates tests for proprietary datasets to evaluate various state-of-the-art models including TAPAS, TAPEX, and CHATGPT.

## 1 Introduction

Table Representation Learning (TRL) models are getting increasing attention for their ability to support various NLP downstream tasks involving tabular data (Badaro et al., 2023; Dong et al., 2022). Such models are built using large open-domain datasets during pre-training and can then be fine-tuned with labelled examples for the target task.

**Motivation.** In most settings, companies aim at adopting a pre-trained model to reduce costs. However, despite the abundance of such methods, it is still challenging to select and manage a TRL model for proprietary data. Models use different pre-training tasks and datasets, work under different assumptions, and the top performing model for an existing benchmark is not necessarily the best one when tested on proprietary data. Indeed, the performance of models fine-tuned on benchmark data is not necessarily replicable on proprietary data.

In a corporate setting, there are at least three scenarios where a given TRL model needs to be evaluated against proprietary datasets:

- Comparison: Compare TRL models fine-tuned on private examples to see which one performs best.
- Validation: As crafting examples is expensive, verify when the quality meets the requirements.
- Maintenance: Fine-tuned models need to be re-calibrated to avoid data and conceptual shifting, continuous evaluation helps the identification of this issue.

**Challenges.** Supporting these scenarios with accurate evaluation is a not obvious task. For the first scenario, consider an engineer who is selecting the TRL model for a Question Answering (QA) task

37th Conference on Neural Information Processing Systems (NeurIPS 2023) Track on Datasets and Benchmarks.

over their enterprise tabular data. Several options are available and they start assessing the models for a zero-shot setting, where the model is used as-is. As labelled examples are not available, the engineer has to craft some test data to assess how the model is performing on their proprietary tables, e.g., write pairs of questions in natural language (NL) and the expected data results.

The task above has a cost that increases with the number of tables and tests crafted by the engineer. Moreover, the process above applies for all the target tasks that the engineer wants to support. For example, they would have to repeat the exercise for the same model and datasets to evaluate its suitability for Semantic Parsing (SP, aka Text2Sql), where, given a question in NL and a table schema, the model returns a SQL query.

Creating tests is only half of the story. Test results must be evaluated, i.e., the data in the QA model's output $D_M$ should "match" the expected output manually crafted in the test $D_T$. A simple equality test fails short, as the tuples (or values) in $D_M$ can be in different order w.r.t. those in $D_T$. Different order for records and attributes does not matter in the relational model, so the test should be independent from this difference. While existing systems are evaluated with a naive accuracy metric that handles these issues, many problems that make this matching hard are ignored. First, they do not test results for relational data integrity, i.e., if tuple and attribute relationships are satisfied. Second, consider tuples in $D_M$ that are similar to the expected ones in $D_T$ except null values, or that differ in the order when the question requires that results are sorted - these issues are ignored by existing metrics. The same observations apply for the Semantic Parsing task, as it is a common practice to evaluate the produced SQL scripts on their results once executed (Li et al., 2023).

**Automatic Assessment.** QATCH (Query-Aided TRL Checklist) addresses the challenges in evaluating TRL models for proprietary data over SQL-centric tasks. QATCH tests TRL models by automatically *generating* and *evaluating* testing checklists of varying complexity for QA and SP.

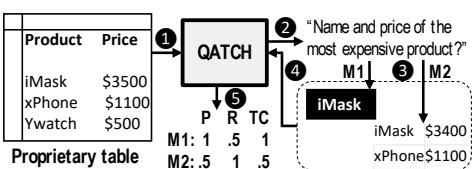

Figure 1: QATCH takes a table as input and returns metrics for TRL models M1 and M2.

Given a proprietary table, such as the one in Figure 1 (step ❶), QATCH produces questions in NL such as the one in the figure (step ❷). The questions are passed, together with the input table, to the TRL models to evaluate (step ❸). The models' output is then consumed by our tool (step ❹) to produce an assessment of the results (step ❺).

At the heart of QATCH, a query generation algorithm crafts tests covering a comprehensive range of features enabled by the expressive power of SQL. Our objective is to assess the capability of the models concerning question complexity. To achieve this, we generate the corresponding NL questions using templates designed to maintain clarity and precision. This approach ensures that we do not inadvertently test the models' capacity to handle complex textual content, as our primary focus is on their performance in relation to question complexity.

QATCH also introduces cross-task performance metrics that evaluate the models' results while accounting for the nuances in the comparison of tabular outputs. To address the complexities in evaluating model outputs, our metrics consider numerous factors when comparing databases, including the cardinality of the results, presence of null values, missing or extra cell values, and sorting when required by the NL questions. By taking these aspects into account, QATCH provides a robust and insightful evaluation of model performance.

Our work differs from existing rigid benchmarks, such as Spider (Yu et al., 2018), which cannot capture how well a model performs on proprietary data. Instead, QATCH supports end-users in corporate settings by facilitating decisions on TRL model comparison, validation and maintenance.

Our contributions are summarized as follows:

- We present QATCH, a toolbox for automatically generating and evaluating test checklists tailored to proprietary data and SQL-centric tasks. Code, data, and results are available at `https://github.com/spapicchio/QATCH`.
- We exploit a SQL query generator to craft a testing checklist for multiple TRL models and two established tasks, i.e., Question Answering and Semantic Parsing.

- We introduce robust performance metrics that capture the nuances in comparing model outputs, such as cardinality, null values, missing or extra cell values, and sorting requirements.
- By executing tests of increasing complexity over four TRL models and 8 proprietary datasets, we report insights into the models' capacity to handle two SQL-centric tasks on tabular data.

## 2  Related work

**Table Representation Learning.** Table Representation Learning (TRL) refers to the process of developing and training neural models to capture the underlying structure, semantics, and relationships in tabular data. TRL is used for tasks both in natural language processing (NLP) and data management. Such models support data-driven systems that surpass the limitations of traditional declarative specifications based on first-order logic and SQL.

Examples of tasks that use these models include answering questions in natural language (Katsogiannis-Meimarakis and Koutrika, 2021; Herzig et al., 2020; Liu et al., 2021), fact-checking (Chen et al., 2020; Yang and Zhu, 2021; Aly et al., 2021), semantic parsing (Yin et al., 2020; Yu et al., 2021), table retrieval (Pan et al., 2021; Kostić et al., 2021; Glass et al., 2021), table comprehension (Suhara et al., 2021; Du et al., 2021), and table content prediction (Deng et al., 2020; Iida et al., 2021). These models are built with different architectures (encoder only, decoder only, encoder+decoder), but our approach is agnostic to this aspect and evaluates any model that can satisfy the input/output requirements of a SQL-centric task, i.e., the output may be obtained with a SQL query. In this work, we focus on two tasks that satisfy these requirements: Question Answering (QA) and Semantic Parsing (SP).

*Question Answering*. In the context of free text, QA aims to retrieve passages containing the answer to a given question. In the tabular data setting, QA involves returning the cells that answer a given query, with the input consisting of a question and a table (Herzig et al., 2020; Liu et al., 2021). There are two levels of complexity in tabular QA tasks. Simple QA focuses on lookup queries on tables, while more complex QA tasks require aggregation operations and numerical reasoning.

*Semantic Parsing*. SP in the tabular data setting involves generating a declarative query in SQL over the table's schema, given a question and a table as input (Yin et al., 2020; Liu et al., 2021; Yu et al., 2021; Li et al., 2023). The purpose of SP is to retrieve the answer to the question by producing an interpretable query rather than directly obtaining the answer. Unlike QA, where the focus is on finding the answer cells, SP emphasizes the generation of a structured query.

Other tasks, such as Table Metadata Prediction (TMP) and Table Content Population (TCP), are also widely covered by TRL models, but they do not follow the task requirements. TMP focuses on predicting inter-table metadata, such as column types, headers, cell types, and table types, as well as intra-table relationships, such as equivalence between columns and entity linking/resolution (Deng et al., 2020; Cappuzzo et al., 2020). Meanwhile, TCP addresses the recovery of corrupted cell content and aims to impute missing cell values in an input table (Iida et al., 2021). This serves to enhance and improve the consistency of table information. However, those are traditional DB problems over tabular data and there exist some relevant benchmark solutions relying on metadata creation (Arocena et al., 2016).

**Previous Benchmarking Approaches.** Several benchmarks have been developed for QA and SP models for measuring model performance on fixed datasets. Examples include those closer to QA (Pasupat and Liang, 2015; Chen et al., 2020) and those for SP (Yu et al., 2018; Gkini et al., 2021). WikiTableQuestions is designed for evaluating table comprehension tasks, with a dataset consisting of questions about tables from Wikipedia. TabFact focuses on fact-checking and comprises statements about tables that can be labeled as "true" or "false". Spider, and similar efforts, are corpora of databases, each with a set of SQL queries to test SP. For each query, they provide the paraphrases of the query as a NL question. These datasets have played a role in advancing TRL models across various tasks. However, they are inherently limited due to their fixed set of crafted examples and lack test generation for different TRL models given only the proprietary datasets.

Moreover, previous approaches have relied on simple accuracy metrics, which test if the produced values are included in the ground truth. This "execution accuracy" is adopted also for SP as it is more precise than just comparing SQL scripts (Li et al., 2023; Yu et al., 2018). However, this metric fails to

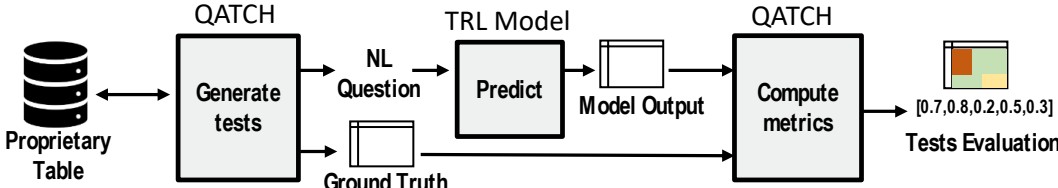

Figure 2: Workflow of QATCH: given proprietary relational tables as input, it generates tests for Question Answering and Semantic Parsing; tests are executed on a given TRL model; the evaluation metrics are computed between predictions (model output) and ground-truth results.

capture nuances in comparing data instances. For example, it does not measure whether values from the same tuple in the ground truth output appear in the same manner in the model output. As a result, this simplistic metric overlooks aspects of TRL models' performance and motivates our proposal.

In terms of test generation, our work took inspiration from an existing effort in measuring the quality of traditional text language models (Ribeiro et al., 2020). Unlike QATCH, Ribeiro et al. (2020) focuses on behavioral testing of NLP models across multiple tasks. Conversely, we leverage the expressive power of SQL to evaluate the test model outcomes directly on tabular data.

## 3 The QATCH Toolbox

QATCH's automatically generates and evaluates test checklists on TRL models based on the three-step process depicted in Figure 2.

1. QATCH-*Generate*. It generates a set of queries tailored to proprietary data. For each query it formulates both the SQL declaration, its free-text version, and the expected ground truth consisting of table instances. The SQL declaration expresses the logical complexity of the query and reflects the presence/absence of specific features peculiar to relational data model.
2. *TRL Model Prediction*. It processes the tests for various TRL models and tasks. The toolbox supports alternative TRL models for both the QA and the SP tasks.
3. QATCH-*Evaluate*. It evaluates the models outputs according to a set of cross-task performance metrics. Separately for each model, it provides end-users with the results for a testing checklist to gain insights into TRL models' strengths and weaknesses.

We describe the tasks covered in the toolbox next. We then explain the test generation. Finally, we discuss the performance metrics.

### 3.1 Question Answering and Semantic Parsing based on TRL models

Question Answering aims at retrieving the answer A to a given question Q, where both Q and A are expressed in natural language. Here we focus on retrieving the content necessary to formulate the answer from a proprietary relational table $T$ encoded by a TRL model M.

We specifically address two sub-tasks tailored to Table Representation Learning Models, i.e., Question Answering from TRLs (QA-TRL, in short) and Semantic Parsing based on TRLs (SP-TRLs).

*Question Answering from TRLs*. Let $T$ be a relational table and let $S_T$ and $I_T$ be $T$'s schema and instance, respectively. Executing a SQL declaration *SQL* on $T$ entails retrieving a (ground truth) result $R_{SQL}$ derived from $I_T$:

$$R_{SQL} = \text{QUERY}(SQL, S_T, I_T) \tag{1}$$

In QA-TRL we do not query the original table $T$, but rather the TRL model M that encodes $T$. This allows us to pose the query in natural language and obtain the result directly from the model.

Let $SQL_Q$ be a SQL declaration and let Q be one of its free-text reformulations, i.e., a question expressed in natural language that is semantically equivalent to $SQL_Q$. The goal of QA-TRL is to ask question Q on M (which encodes $T$) to retrieve a result $R_Q$

$$R_Q = \text{QA-TRL}(M, Q) \tag{2}$$

| Category | SQL declaration | Free-Text question |
|---|---|---|
| Project | SELECT $\{c_1, \ldots, c_n\}$ FROM $\{T\}$ | Show $\{c_1, \ldots, c_n\}$ in table $\{T\}$ |
| Distinct | SELECT DISTINCT $\{c_1, \ldots, c_n\}$ FROM $\{T\}$ | Show the different $\{c_1, \ldots, c_n\}$ in table $\{T\}$ |
| Select | SELECT * FROM $\{T\}$ WHERE $\{c_i\}$ $\{op\}$ $\{val\}$ | Show data of table $\{t\}$ where $\{c_i\}\{op\}\{val\}$ |
| Order by | SELECT * FROM $\{T\}$ ORDER BY $\{c_i\}$ $\{ord\}$ | Show data for table $\{T\}$ in $\{ord\}$ order by $\{c_i\}$ |

Table 1: Templates for queries in SQL and natural language. $T$ is the target relational table. $c_i \in S_T$ $(1 \leq i \leq n)$ is an attribute of the $T$'s schema, *op* $(=, !=, >, <, \geq, \leq)$ is a logical operator, VAL$_i$ is an arbitrary value for attribute $c_i$ occurring in I$_T$. *ord* is the order of visualization of the tuples in the output (i.e., ascending or descending).

s.t. $R_Q$ is equivalent (or mostly similar, in the worst case) to the expected outcome $R_{SQL}$ from $T$.

*Semantic Parsing using TRLs*. Given a free-text question Q and a relational table $T$, the goal of SP is to map Q to the corresponding SQL declaration SQL$_Q$.

In SP-TRL we leverage the TRL model $M$, which encodes the table schema $S_T$, to perform the Text2SQL transformation.

$$\text{SQL}_Q = \text{SP-TRL}(Q, M) \tag{3}$$

*Performance Evaluation*. Testing TRL models for QA-TRL and SP-TRL provides *complementary information* about their ability to encode complex instance- and schema-level relations holding in tabular data. Instead of verifying the syntactic overlap between free-text questions and SQL declarations, we evaluate results' consistency at the tuple level. Specifically, in QA-TRL we measure how similar the result $R_Q$ returned by the query $Q$ and the expected outcome $R_{SQL}$ are in terms of tuple characteristics and cell values. Similarly, in SP-TRL we compare the tuples retrieved by the execution of the output query SQL$_Q$ on $T$ with those contained in the expected outcome $R_{SQL}$.

QATCH *Key Steps*. QATCH automatically generates SQL declarations SQL$_Q$ and free-text questions Q from a proprietary table $T$. Based on the table schema and instances, a query generator first returns a set $\mathcal{S}$ of SQL declarations of varying complexity.

$$\mathcal{S} = \text{QUERYGENERATOR}(T) \tag{4}$$

Then, for every query SQL$_Q \in \mathcal{S}$, QATCH generates a *template* corresponding to one of its free-text versions.

$$Q = \text{GENERATETEMPLATE}(\text{SQL}_Q) \tag{5}$$

The separation of queries and questions allows us to disentangle the characteristics of the output from the linguistic capabilities of the models. Free-text questions and SQL declarations are used to test the TRL models on the QA-TRL an SP-TRL tasks.

### 3.2 Automatic generation of testing checklists with SQL queries

Given a relational table $T$, QATCH relies on a SQL query generator to produce queries of varying logical complexity for a TRL model's assessment. Each test query consists of a triple $\langle Q, \text{SQL}_Q, I_Q^{gt} \rangle$, where SQL$_Q$ is the input query formulated as a SQL declaration, Q is one of the possible free-text reformulation of the question, and $I_Q^{gt}$ is the corresponding ground truth.

Table 1 enumerates the templates used to generate SQL$_Q$ first and then Q based on proprietary data. They encompass SQL projections on the table schema (Project), selections on the table instance (Select), and more complex cardinality-based and sorting criteria at the tuple level (Distinct and Order by)[1]. The variable in the templates are automatically filled up by the tool according to the values in the schema and active domain on the given $T$.

The SQL generator automatically crafts multiple versions of the input tests, each one incorporating a different feature. For example, for the Project template, QATCH explores the following three cases: (1) SELECT ALL, which selects all the attributes in the table schema; (2) SELECT-RANDOM-COL,

---

[1] Since existing QA models work on single tables, we currently disregard Join operations.

which selects $n$ random subsets of attributes, each subset of cardinality $1,\ldots,n$, where $n$ is the number of attributes in the table schema; (3) `SELECT-ADD-COL`, which starts from a fixed partition $P_S \in S_T$ of the table schema and incrementally adds one random attribute at a time. The SQL generator also mixes categories to create more complex queries, e.g., using projection and selection in the same test.

The motivation behind the proposed templates is to examine the TRL model's capability to handle various algebraic operations that may be directly or indirectly applied to the relational schema (based on the targeted task). Our goal is not to stress the complexity of the linguistic expression, such as ambiguity (Veltri et al., 2023), bur rather to analyze the *logical complexity of the question according to the relational data model*.

### 3.3 Cross-task performance metrics

QATCH summarizes the outcomes of the tests with a set of cross-task performance metrics defined on the comparison of the model output against the expected ground truth. The comparison reflects the similarity between the corresponding instances and the ability of TRL models to handle specific data issues. Specifically, given $I_Q^{cell}$ and $I_Q^{cell,gt}$ as the sets of distinct pairs respectively occurring in $I_Q$ and $I_Q^{gt}$, it supports the following metrics:

**Cell Precision**. The fraction of table cells $cell_i$ in the output instances that are relevant to the input query. The higher is the score, the more predicted elements are in the target. However, it does not measure how many target cells are in the prediction (measured by cell recall).

$$\text{C-PR} = \frac{|\{cell_i | cell_i \in I_Q^{cell} \cap I_Q^{cell,gt}\}|}{|\{cell_i | cell_i \in I_Q^{cell}\}|}$$

**Cell Recall**. The fraction of table cells that are relevant to the input query that are successfully retrieved. The higher is the score, the more target cells are in the prediction. It does not measure how many prediction cells are in the target (measured by cell precision).

$$\text{C-REC} = \frac{|\{cell_i | cell_i \in I_Q^{cell} \cap I_Q^{cell,gt}\}|}{|\{cell_i | cell_i \in I_Q^{cell,gt}\}|}$$

**Tuple constraint**. The fraction of ground truth tuples in the query output. It is one if the expected and produced outputs have both the same schema, the same cardinality and the same cell values, zero otherwise. However this hard constraint does not capture all the cases (see example below).

$$\textit{T-Cons} = \frac{|I_Q \cap I_Q^{gt}|}{|I_Q^{gt}|}$$

**Tuple cardinality**. The ratio of output and ground truth cardinality. It is a "softer" constraint w.r.t. the tuple constraint since it does not consider neither the schema nor the cell values. However, it has to be analysed with cell precision and cell recall to be meaningful.

$$\text{T-CARD} = \frac{|I_Q|}{|I_Q^{gt}|}$$

**Tuple Order**: the Spearman rank correlation coefficient (Katsaounis, 2003) between the vector representations of the ranked lists of returned and ground truth tuples. This non-parametric test is computed only for queries with the Order-By clause.

Figure 3 shows examples of the performance metrics on a toy dataset. Given the ground truth result (target) with three tuples over two attributes, we report the metric values for five predictions, coming either from a QA or from the execution of a query in SP. Notice that:

• Tuple constraint/cardinality and cell-precision/recall range between 0 (no matches) and 1 (all matches), while the tuple order ranges between 0 (opposite rank) and 1 (same rank).

| | TARGET | [[apple, red], [pear, green], [banana, yellow]] | CELL PRECISION | CELL RECALL | TUPLE CONSTRAINT | TUPLE CARDINALITY | TUPLE ORDER |
|---|---|---|---|---|---|---|---|
| PREDICTION 1 | | [[apple, red], [banana, yellow]] | 1.0 | 4/6 | 2/3 | 2/3 | 1.0 |
| PREDICTION 2 | | [[apple], [pear], [banana]] | 1.0 | 0.5 | 0 | 1.0 | 0.5 |
| PREDICTION 3 | | [[grape, blue], [pineapple, yellow]] | 1/4 | 1/6 | 0 | 2/3 | 0.5 |
| PREDICTION 4 | | [[pear, green], [apple, red], [banana, yellow], [peach]] | 6/7 | 1.0 | 1.0 | 3/4 | 0.75 |
| PREDICTION 5 | | [[pear, red], [apple, yellow], [banana, green]] | 1.0 | 1.0 | 0 | 1.0 | 0.5 |

Figure 3: QATCH's metrics are computed between the model output (prediction) and expected ground-truth results (target). The target is the answer of the NL question "*Show me all the data*" over a table with three tuples and two attributes.

- The tuple constraint ignores partial tuple matches (e.g., [APPLE, RED] is different from [APPLE] in Prediction 2), which is captured by cell recall.
- When the number of expected tuples is zero, tuple cardinality takes either value zero, if the number of predicted tuples is greater than zero, or one, if no tuples are predicted.

To explain the rationale behind the proposed metrics, let us consider the following target values (a1, b1, c1). The first prediction, denoted by "output 1", is composed of two tuples (a1, b1, c1), (a1, b1, c1). Instead, the second prediction, denoted by "output 2", has the following two tuples: (a1, b1), (c1). "Output 1" has cardinality two instead of one whereas "output 2" has an incorrect schema. In both cases the tuple constraint returns zero even if part of the output cells match. Notice also that even if the outputs' cardinality is the same cell precision and recall show relevant differences in the returned values.

Our Cell Precision is equivalent to the Execution Accuracy reported in most QA and SP papers. We remark that while informative, this single metric does not capture cases where the output is incorrect, such as Prediction 5 in Figure 3.

## 4 Experimental Evaluation

**TRL models.** We test six TRL models and one Large Language Model, i.e., CHATGPT (OpenAI, 2023). For QA-TRL, we report for TAPAS (Herzig et al., 2020), TAPEX (Liu et al., 2021), and OMNITAB (Jiang et al., 2022). TAPAS leverages the transformer architecture to pre-train on large-scale datasets and fine-tune to specific tasks using labeled examples. TAPEX exploits the idea of learning in the pre-training a neural SQL executor over a synthetic corpus of SQL queries and their execution outputs. OMNITAB exploits a pretraining approach that uses both natural and synthetic data to learn reasoning over multiple table elementes. We use TAPAS and TAPEX fine-tuned on the WTQ dataset for QA.

For SP-TRL, we report for RESDSQL (Li et al., 2023), GAP (Shi et al., 2021), UNIFIEDSKG (Xie et al., 2022). RESDSQL is based on a seq2seq architecture with a ranking-enhanced encoding and skeleton-aware decoding framework. UNIFIEDSKG implements the encoder-decoder (text-to-text) model based on T5. We use UNIFIEDSKG and RESDSQL with the T5 large setting. GAP employs generative models to create pre-training data, enabling the simultaneous learning of natural language utterances and table schemas.

CHATGPT is a language model employing a decoder-only transformer setup; by leveraging a chat interface, it allows to specify a range of tasks (including QA and SP) with the in-context adaptation technique. This technique leverages the model's ability to understand and consider the immediate context of a conversation or text. Further details to reproduce the results can be found in the appendix.

| Category | Table Name | # rows | # categorical cols | # numerical cols | Example cols |
|----------|-----------|--------|--------------------|-----------------|--------------|
| ECOMMERCE | Sales-transactions | 500k | 5 | 3 | ProductNo, Date |
| | Fitness-trackers | 565 | 8 | 3 | Brand Name, Display |
| FINANCE | Account-fraud | 1M | 4 | 26 | DaysSinceRequest, Velocity6h |
| | Late-payment | 2466 | 6 | 6 | InvoiceDate, Disputed |
| MEDICINE | Heart-attack | 303 | 1 | 11 | # trtbps, # oldpeak |
| | Breast-cancer | 686 | 5 | 6 | pgr, rfstime |
| MISC | Adult-census | 32.6k | 9 | 6 | education, fnlwgt |
| | Mushrooms | 8.1k | 23 | 0 | cap-shape, ring-type |

Table 2: Information for the proprietary tables used in the experiments.

**Datasets.** We generate tests for 8 proprietary tables that we have selected from Kaggle to maximize variety both in terms of content (four categories) and in terms of data properties (different sizes and arity), as detailed in Table 2. These datasets are available online, thus possibly "seen" by the models, but are not available for question answering and semantic parsing tasks, i.e. they do not come with natural language questions or queries. The 8 tables are used in this paper as a sample to show the benefit of automatic test generation with QATCH, we are not suggesting to use them as a new corpus with a fixed set of questions and tables. We also report results for the widely used Spider benchmark, as this is the standard for QA and SP tasks evaluation, using its tables and its questions. This comparison aims to emphasize the discrepancies between the quality results obtained using Spider and the practical application of models on proprietary tables. More details on the data processing phase can be found in the appendix.

As models are limited in the input, for every table we sample a subset of rows and attributes that can be executed within the model context (e.g., about 4k tokens for CHATGPT).

| Category | Model | Cell precision | Cell recall | Tuple cardinality | Tuple constraint | Tuple order | Avg |
|----------|-------|----------------|-------------|-------------------|------------------|-------------|-----|
| | PROPRIETARY DATA | | | | | | |
| ECOMMERCE | TAPAS-LARGE-WTQ | **0.71** | 0.12 | **0.53** | 0.05 | 0.33 | **0.35** |
| | TAPEX-LARGE-WTQ | 0.40 | 0.06 | 0.18 | 0.01 | 0.40 | 0.21 |
| | OMNITAB | 0.20 | 0.01 | 0.14 | 0.00 | **0.50** | 0.17 |
| | CHATGPT 3.5 | 0.44 | **0.24** | 0.20 | **0.10** | 0.42 | 0.28 |
| FINANCE | TAPAS-LARGE-WTQ | **0.72** | 0.12 | **0.48** | 0.05 | 0.38 | 0.35 |
| | TAPEX-LARGE-WTQ | 0.52 | 0.06 | 0.16 | 0.01 | 0.48 | 0.25 |
| | OMNITAB | 0.30 | 0.02 | 0.13 | 0.00 | **0.50** | 0.19 |
| | CHATGPT 3.5 | 0.71 | **0.52** | 0.38 | **0.21** | 0.48 | **0.46** |
| MEDICINE | TAPAS-LARGE-WTQ | 0.72 | 0.16 | **0.57** | 0.09 | 0.34 | 0.38 |
| | TAPEX-LARGE-WTQ | 0.37 | 0.04 | 0.15 | 0.0 | 0.44 | 0.20 |
| | OMNITAB | 0.29 | 0.01 | 0.12 | 0.0 | 0.50 | 0.18 |
| | CHATGPT 3.5 | **0.77** | **0.46** | 0.22 | **0.12** | **0.70** | **0.45** |
| MISCELLANEOUS | TAPAS-LARGE-WTQ | 0.67 | 0.12 | 0.34 | 0.04 | 0.29 | 0.29 |
| | TAPEX-LARGE-WTQ | 0.48 | 0.10 | 0.25 | 0.01 | 0.44 | 0.26 |
| | OMNITAB | 0.12 | 0.02 | 0.13 | 0.01 | **0.50** | 0.17 |
| | CHATGPT 3.5 | **0.76** | **0.67** | **0.36** | **0.16** | 0.50 | **0.49** |
| | EXISTING BENCHMARK DATA | | | | | | |
| Spider | TAPAS-LARGE-WTQ | 0.64 | 0.42 | 0.53 | 0.30 | 0.64 | 0.51 |
| | TAPEX-LARGE-WTQ | 0.62 | 0.45 | 0.54 | 0.21 | 0.51 | 0.47 |
| | OMNITAB | 0.30 | 0.24 | 0.53 | 0.23 | 0.52 | 0.36 |
| | CHATGPT 3.5 | **0.74** | **0.77** | **0.86** | **0.66** | **0.75** | **0.76** |

Table 3: Results for QA-TRL models: average of all tests on multiple tables. ChatGPT version: ChatGPT 3.5-turbo-0613

| Category | Model | Cell precision | Cell recall | Tuple cardinality | Tuple constraint | Tuple order | Avg |
|---|---|---|---|---|---|---|---|
| | | PROPRIETARY DATA | | | | | |
| ECOMMERCE | RESDSQL | 0.91 | 0.89 | 0.92 | 0.81 | **1.00** | 0.90 |
| | GAP | 0.84 | 0.80 | 0.81 | 0.73 | 0.97 | 0.83 |
| | UNIFIEDSKG | 0.71 | 0.71 | 0.69 | 0.69 | **1.00** | 0.76 |
| | CHATGPT 3.5 | **0.98** | **0.98** | **0.99** | **0.95** | **1.00** | **0.98** |
| FINANCE | RESDSQL | 0.90 | 0.87 | 0.95 | 0.77 | **1.00** | 0.90 |
| | GAP | 0.79 | 0.78 | 0.76 | 0.74 | **1.00** | 0.81 |
| | UNIFIEDSKG | 0.79 | 0.76 | 0.74 | 0.67 | 0.98 | 0.79 |
| | CHATGPT 3.5 | **0.96** | **0.96** | **0.99** | **0.90** | **1.00** | **0.96** |
| MEDICINE | RESDSQL | 0.86 | 0.75 | 0.94 | 0.67 | 0.95 | 0.83 |
| | GAP | 0.77 | 0.73 | 0.73 | 0.67 | 0.59 | 0.70 |
| | UNIFIEDSKG | 0.72 | 0.69 | 0.70 | 0.66 | 0.95 | 0.74 |
| | CHATGPT 3.5 | **1.00** | **1.00** | **0.98** | **0.99** | **1.00** | **0.99** |
| MISCELLANEOUS | RESDSQL | 0.94 | 0.90 | 0.90 | 0.77 | **1.00** | 0.90 |
| | GAP | 0.82 | 0.78 | 0.73 | 0.69 | **1.00** | 0.80 |
| | UNIFIEDSKG | 0.74 | 0.69 | 0.68 | 0.59 | 0.98 | 0.73 |
| | CHATGPT 3.5 | **0.98** | **0.98** | **0.98** | **0.91** | **1.00** | **0.97** |
| | | EXISTING BENCHMARK DATA | | | | | |
| Spider DEV | RESDSQL | 0.93 | 0.93 | **0.97** | 0.84 | 0.99 | 0.93 |
| | GAP | **0.95** | 0.95 | 0.96 | 0.91 | 0.96 | **0.95** |
| | UNIFIEDSKG | 0.81 | 0.82 | 0.82 | 0.80 | **1.00** | 0.85 |
| | CHATGPT 3.5 | 0.93 | **0.96** | **0.97** | **0.92** | 0.90 | 0.94 |
| Spider TRAIN | CHATGPT 3.5 | 0.90 | 0.92 | 0.92 | 0.88 | 0.97 | 0.92 |

Table 4: Results for SP-TRL models: average of all tests on multiple tables. CHATGPT version: CHATGPT 3.5-turbo-0613. UNIFIEDSKG and RESDSQL evaluated with the T5 large.

**Results.** Tables 3 and 4 show the results for QA-TRL and SP-TRL, respectively. All metric results are averaged over all tests and over tables grouped by category; we report detailed results for every dataset and every test category in the appendix.

For the QA-TRL task, Table 3 reports the results achieved on the proprietary tables and on the Spider benchmark to enable a comparison between the two groups. All models show promising results in terms of Cell precision, which is the metric closest to the one used in previous papers. However, the other metrics show lower values, with all models struggling in preserving the intra-tuple value relations in their output (i.e., low Tuple constraints). Also, in almost all experiments the models tend to remove duplicates, even when the `Distinct` clause is not present. This is evident with the low scores for Cell recall and Tuple Cardinality. Finally, none of the models is able to return results according to the `Order by` requirements (i.e., low Tuple order scores).

Results for the SP task in Table 4 show that, in general, the TRL models performs much better in this task. The most problematic aspect is the preservation of the tuple structure for the returned values (Tuple constraint).

Tests can be generated for a specific dataset using QATCH in just a few seconds. The testing times varies based on the data, particularly the size of the table. For instance, comparing a prediction and a ground truth containing 1000 tuples and 20 attributes takes less than a second.

As we do not control the infrastructure of OpenAI, we do not report exact execution times for CHATGPT. On average, it takes a few seconds to execute one test, but the distribution for this metrics is skewed as it heavily depends on the server workload.

**Discussion.** The results show that the tests generated by QATCH and its metrics enable detailed evaluation of the models. A clear message is that there is no single model that works best for every table and every metric. In some datasets, TAPAS has the best performance in terms of cell precision and recall, but CHATGPT is the best model in terms of tuple cardinality and tuple constraint. The

lower precision for CHATGPT in some cases is due to the hallucination problem with the decoder architecture.

Another take-away is that questions and tables in Spider do not tell the full story about model performance. Even a small selection of 8 tables show that the performance for established TRL models can drop dramatically on propriety data in the QA task.

Looking into the details of the failed tests, it is easy to spot the limitations of the models. Cell precision is lowered by select queries with more than one column in the output. The models tend to return only one attribute even in this setting.

In some cases, Tuple cardinality is lower than Cell recall, because the latter is over a set. For some queries, models return DISTINCT by default. Most of the questions in Spider return one tuple only, thus inflating the aggregate result and hiding the problem with more output attributes. For the same reason, tuple constraint scores are low because in many cases the models do not return tuple structures, but mostly separated cells. We again observe low results for proprietary data with the higher ones for Spider.

We can conclude that the qualitative performance of a TRL model depends on the target task at hand, but also on the tabular content in terms of data domain, size, schema, and data quality.

## 5 Conclusion and Future Work

We presented QATCH, a testing tool for TRL models. It provides end-users with a flexible and adaptive solution for the automatic assessment of models' performance on proprietary data. The key goal is to avoid trusting the results achieved on the existing benchmarks across different tasks, as they are not always replicable on proprietary data. By leveraging the expressive power of the SQL language, we first generate queries of varying complexity on proprietary table and then convert them in natural language. Thus, we specifically assess the model capability to address complex queries rather than the linguistic properties of questions and answers. Results show that (1) existing questions in benchmarks do not capture important properties of custom tables, (2) popular metrics fail short in measuring the quality of the models' output.

As future research agenda, we plan to investigate the following research directions: (1) The extension of QATCH towards the automatic assessment of additional models (e.g., Bard (Google Inc., 2023), LLama2 (Touvron et al., 2023), Falcon (Penedo et al., 2023)), tasks (e.g., fact checking (Guo et al., 2022; Nakov et al., 2021), querying LLMs (Saeed et al., 2024; Urban et al., 2023)), and more complex queries (e.g., GROUP BY clauses, nested queries). (2) The adoption of Generative Language Models (e.g., (OpenAI, 2023)) to automatically generate templates in Natural Language from the SQL declarations. (3) The use of TRL models to test inter-relational constraints, such as joins and referential constraints, among proprietary tables.

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

# A Authors' statements

## A.1 Code license and legal compliance

The source code of our project is released under the **Apache-2.0** license. All the authors are aware of the legal implications and gave their explicit consent to publish the paper and the code.

## A.2 Reproducibility

The code and the related documentation is available at: `https://github.com/spapicchio/QATCH`. The source code is annotated in compliance with the ML reproducibility checklist provided by NeurIPS 2023 organizers and available at `https://github.com/paperswithcode/releasing-research-code`. We also adhere to the official guidelines for the specification of the code dependencies. Notice that TRL models are referenced but not directly included in the main repository as we exclusively rely on third-party models.

## A.3 Long-term preservation

We guarantee that the repository, including the project code and annotations, will be maintained in the long term. Any potential issues that will be raised by end-users will be taken into consideration.

## A.4 Privacy

In this work, we released a new testing toolbox for proprietary data. We neither released a new publicly available datasets nor involved humans in the evaluation loop.

While using QATCH the input data, the tests, and the expected and achieved outcomes always remain under the full end-user's control as long as TRL models are run on premises (no external APIs). Thus, QATCH allows a secure testing of proprietary tables by running models and tests on a local machine.

For the sake of completeness, QATCH also supports an external QA model, i.e., OpenAI ChatGPT 3.5. While using it, end-users must be aware of the fact that their own data are shared with private subjects in compliance with OpenAI terms and policies (`https://openai.com/policies`).

## A.5 Ethics

There are no particular ethical implications of our work. Every ethical issue related to the tested data is under the liability of the data owner.

## A.6 Bias and fairness

The QATCH test generation module randomly picks existing content and/or metadata from the input tables. While running the tests, we recommend to also vary the seed of the random generator to maximize the reliability of the tests' outcomes. Mitigating the eventual presence of bias in the input data and testing TRL model fairness are both out of scope of the present work.

## A.7 Impact of the environment

Testing TRL models on proprietary data prevents the use of expensive and resource demanding cloud services. Since QATCH allows end-users to optimize the computational efforts by choosing the right model to use. Concerning that point, its use can be deemed as an example of *green data mining* Schneider et al. (2023).

# B Additional details on data, models, and tests

**Tables' description.** Here we describe the tables used in the experimental evaluation. We collected results on eight proprietary tables and on one benchmark dataset established for TRL model evaluation Yu et al. (2018).

We considered proprietary tables of various domains (i.e., E-Commerce, Finance, Medicine, and Miscellaneous) and characteristics (e.g., tuple cardinality from few hundreds to more than 1 million, table dimensionality from 8 to 30) to ensure the robustness of the empirical outcomes. The detailed statistics are given in Table 2 of the main paper.

In Table 5 we reported similar statistics for the Spider benchmark dataset. These statistics were not included in the main paper due to the lack of space.

**Proprietary data preprocessing.** QATCH supports standard text pre-processing procedures for the table content and metadata. For example, the Mushrooms table contains only alphabetical letters thus conveying limited information. We applied a text reconciliation step, where, for instance, in the `Heart-attack` table the attribute value *cp* is replaced by the (more meaningful) expression *chest pain*.

**Spider preprocessing.** The primary objective of QATCH is to assess the impact of proprietary data on model performance. To achieve this goal, QATCH tests are deliberately designed to explore model behavior in simplified scenarios, avoiding intricacies in both the generated questions and queries. In contrast, Spider, having a different scope, inherently contains a higher level of complexity. To ensure a fair comparison between the proprietary data-driven QATCH tests and the Spider dataset, a preprocessing step is applied for the Spider dataset. Specifically, we excluded queries involving the LIMIT, INTERSECT, UNION, JOIN and EXCEPT operators, as well as those with inner queries.

**Benchmark vs. proprietary data.** The tables included in the Spider benchmark have quite different characteristics compared to the proprietary ones (fewer tuples and dimensions, fewer numerical attributes).

For example, in Table 5) we report the comparison between the Spider tables and the selected proprietary tables. The number of cells per table in Spider is significantly lower in 75% of the cases. This confirms the limited representativeness of the currently used benchmark data for TRL model evaluation and fosters the need of TRL testing tools on proprietary data.

| | | minimum | 25% percentile | 50% percentile | 75% percentile | maximum |
|---|---|---|---|---|---|---|
| Spider tables | # tuples | 0 | 7 | 12 | 15 | 16049 |
| | # attributes | 2 | 4 | 5 | 7 | 24 |
| | # cells | 0 | 30 | 55 | 100 | 112343 |
| Proprietary tables | # tuples | 303 | 656 | 5283 | 149k | 1M |
| | # attributes | 8 | 11 | 12 | 17 | 30 |
| | # cells | 3636 | 7213 | 108k | 1.4M | 30M |

Table 5: Comparison between the Spider and proprietary tables' statistics. It reports the minimum and maximum values, and the 25%, 50%, and 75% percentile over all the Spider tables and the considered proprietary tables.

**Tackling TRL model limitations.** To meet the TRL models' constraints on the maximum number of processed cells, whenever strictly necessary QATCH reduces the input data size by sampling the tables attributes and rows using the same random seed for all tables. To avoid introducing bias in the data, we tested multiple dataset variants consisting of different attribute and tuple selections. More specifically, to address Question Answering we generated all the tests by exclusively considering the table schema, except for the `Select` queries where the tuple-level conditions are generated by considering as representatives the maximum, minimum, or mean attribute values computed separately for each table. The main goal here is to test whether the TRL model can accurately answer queries including arbitrary selection conditions.

The inherent limitations of TRL models to handle larger tables do not influence Semantic Parsing tests, as they directly work at the schema level. Notice that such limitations are independent of the QATCH implementation, but can be conveniently highlighted while running the tests on proprietary data.

| Table name | Selected columns | # of rows Tapas-wtq | # of rows Tapex-wtq | # of rows ChatGPT | # of rows Omnitab |
|---|---|---|---|---|---|
| Sales-transactions link | TransactionNo, ProductNo, Product, Price Quantity, CustomerNo, Date, Country | 60 | 20 | 40 | 20 |
| Fitness-trackers link | Brand Name, Device Type, Original Price, Selling Price, Color, Model Name, Average Battery Life (in days), Rating (Out of 5) Strap Material, Display | 50 | 25 | 30 | 20 |
| Account-fraud link | income, payment_type, has_other_cards, email_is_free, employment_status, housing_status, date_of_birth_distinct_emails_4w, device_os credit_risk_score, session_length_in_minutes | 50 | 25 | 30 | 20 |
| Late-payment link | CustomerID, PaperlessDate, InvoiceNumber, Invoice Date, DueDate, InvoiceAmount, Disputed, PaperlessBill, DaysToSettle, DaysLate | 50 | 20 | 25 | 20 |
| Heart-attack link | age, sex, cp, trtbps, chol, fbs, restecg, caa, thall, output | 45 | 30 | 30 | 20 |
| Breast-cancer link | pid, age, meno, size, grade, nodes, pgr, er, hormon, status | 45 | 30 | 30 | 20 |
| Adult-census link | workclass, education, marital.status, occupation, relationship, race, sex, hours.per.week, native.country, income | 50 | 20 | 30 | 20 |
| Mushrooms link | class, cap-shape, cap-surface, cap-color, bruises, odor, gill-attachment, gill-spacing, gill-size, gill-color | 50 | 25 | 30 | 20 |

Table 6: Datasets information with maximum number of rows accepted for each model in the QA Scenario.

In Table 6 we report the dataset, the list of table attributes, and the maximum number of tuples that each TRL model (and the corresponding tokenizer) manages to process. It is worth noticing that, even though Tapex is based on BART (maximum token limit: 1024) and Tapas is based on BERT (token limit: 512), the number of tuples that Tapas is able to process is much larger. The link to each dataset along with the pre-processing strategies are available in the github repository.

**Performance metrics.** Our evaluation metrics rely on the predicted cell values, making them essential in the SP scenario. To begin with, we employ models to predict SQL queries, which are subsequently executed on tables to retrieve the corresponding predicted cell values. This approach allows us to derive valuable insights from the metric scores associated with the SQL predictions. For instance, a low Tuple constraint metric indicates that the SQL prediction is likely projecting incorrect columns. Instead, a low tuple cardinality metric suggests that the SQL prediction may be selecting incorrect rows or utilizing a distinct operator inappropriately. Table 7 shows three instances of unsuccessful predictions.

**Table linearization.** Using ChatGPT, which processes text sequentially from left to right, necessitates converting tabular data into a linear format for seamless integration. This process, known as linearization, entails transforming structured tabular data into a sequential layout that aligns with ChatGPT's processing method. A common technique involves concatenating cell contents within a row, using specific delimiters for separation. However, a more promising approach is to enhance this by appending column headers to cell content, incorporating structure-aware indicators as described in Badaro et al. (2023). The approach used in this work, illustrated in Figure 4, represents the table as a list of lists, using Python-like syntax. In this representation, each inner list corresponds to a distinct row in the table, with elements indicating values associated with respective columns marked by the

|  |  | Cell precision | Cell recall | Tuple cardinality | Tuple constraint | Tuple order |
|---|---|---|---|---|---|---|
| **Target** | SELECT DISTINCT emailisfree FROM fraud | 0.5 | 1.0 | 0.2 | 0.0 | - |
| **Prediction** | SELECT emailisfree, income FROM fraud |  |  |  |  |  |
| **Target** | SELECT emailisfree FROM fraud ORDER BY emailisfree ASC | 1.0 | 1.0 | 1.0 | 1.0 | 0.0 |
| **Prediction** | SELECT emailisfree FROM fraud ORDER BY emailisfree DESC |  |  |  |  |  |
| **Target** | SELECT * FROM fraud | 1.0 | 0.10 | 1.0 | 0.0 | - |
| **Prediction** | SELECT emailisfree FROM fraud |  |  |  |  |  |

Table 7: Examples of predictions and performance metrics.

Figure 4: Example of the linearization technique used in this study to process structured data with ChatGPT.

structure-aware indicator [H]. This format ensures the preservation of the table's structural integrity and facilitates seamless processing within the ChatGPT framework.

**ChatGPT reproducibility.**   In this study we use ChatGPT 3.5-turbo-0613 with the "in-context" task adaptation technique. In-context learning refers to the ability of models like GPT-3 to consider the immediate context of a conversation or text when generating responses. This means the model does not just look at the current question or prompt, but also the preceding conversation or text. For instance, in-context learning can be seen as a form of few-shot learning where the 'shots' are the previous turns in the conversation. For our study, we pass three examples[2]: (i) the projection of all the attributes (ii) the projection of one column (iii) one aggregate condition.

## C   Additional results for Question Answering

Table 8 compares the performance of the QA models on proprietary data.

By examining the achieved results, we identified the following patterns and trends in the models' performance:

- Tapas and Tapex consistently exhibit higher cell precision but lower cell recall. This indicates that their predictions are often correct yet incomplete.

- For the `Projection` queries, the models understand which columns to include but only return a portion of the expected values.

- All the models fail to return the values in the expected tuple format. This indicates that the models struggle to comprehend the structure of the table.

- Although ChatGPT generally achieves a high overall score, it is not always the best performing model. Specifically, it struggles to cope with the `Sales-transactions` and `Late-payment` tables. Tuple cardinality and Tuple constraint metrics were consistently below 0.50, indicating that the returned outcomes are still far from being optimal.

Table 9 provides additional insights by showcasing the SQL patterns in which ChatGPT performed poorly. It performs fairly good when the number of attributes in the output is limited (e.g., `DISTINCT-SINGLE`). Conversely, it struggles when a larger number of attributes and values need to be considered, especially they mainly include numerical attributes. For example, `Sales-transactions` has 7 out of 8 columns as numerical, while `Late-payment` has 7 out of 10 columns as numerical.

---

[2]More details can be found in the GitHub https://github.com/spapicchio/QATCH/

The same trend is also evident in the improved performances of ChatGPT when dealing with tables in the same category, such as `Fitness-trackers`, which has 6 categorical attributes out of 10, and `Fraud`, which has 7 categorical attributes out of 10 or, even more noticeable, in the `Mushrooms` table where all the 10 attributes are categorical.

| Table category | Table name | Model | Cell precision | Cell recall | Tuple cardinality | Tuple constraint | Tuple order |
|---|---|---|---|---|---|---|---|
| ECOMMERCE | Sales-transactions | TAPAS-WTQ | **0.75** | **0.13** | **0.67** | **0.05** | **0.50** |
| | | TAPEX-WTQ | 0.40 | 0.07 | 0.21 | 0.02 | 0.28 |
| | | OMNITAB | 0.24 | 0.02 | 0.17 | 0.0 | 0.50 |
| | | CHATGPT | 0.15 | 0.03 | 0.05 | 0.02 | 0.0 |
| | Fitness-trackers | TAPAS-WTQ | 0.60 | 0.08 | 0.32 | 0.04 | 0.20 |
| | | TAPEX-WTQ | 0.45 | 0.06 | 0.16 | 0.0 | **0.50** |
| | | OMNITAB | 0.16 | 0.01 | 0.12 | 0.01 | 0.50 |
| | | CHATGPT | **0.67** | **0.46** | **0.29** | **0.27** | 0.44 |
| FINANCE | Account-fraud | TAPAS-WTQ | 0.64 | 0.11 | 0.38 | 0.06 | 0.35 |
| | | TAPEX-WTQ | 0.51 | 0.07 | 0.14 | 0.01 | **0.50** |
| | | OMNITAB | 0.26 | 0.02 | 0.11 | 0.00 | 0.50 |
| | | CHATGPT | **0.68** | **0.61** | **0.42** | **0.17** | 0.48 |
| | Late-payment | TAPAS-WTQ | **0.77** | **0.12** | **0.55** | 0.05 | 0.40 |
| | | TAPEX-WTQ | 0.48 | 0.05 | 0.18 | 0.0 | **0.45** |
| | | OMNITAB | 0.35 | 0.02 | 0.16 | 0.0 | 0.50 |
| | | CHATGPT | 0.13 | 0.05 | 0.04 | **0.03** | 0.0 |
| MEDICINE | Heart-attack | TAPAS-WTQ | 0.60 | 0.13 | **0.45** | 0.08 | 0.23 |
| | | TAPEX-WTQ | 0.31 | 0.05 | 0.16 | 0.01 | 0.38 |
| | | OMNITAB | 0.19 | 0.01 | 0.11 | 0.0 | 0.50 |
| | | CHATGPT | **0.87** | **0.74** | 0.41 | **0.35** | **0.58** |
| | Breast-cancer | TAPAS-WTQ | **0.83** | 0.18 | **0.67** | 0.10 | 0.45 |
| | | TAPEX-WTQ | 0.42 | 0.04 | 0.15 | 0.0 | 0.50 |
| | | OMNITAB | 0.37 | 0.02 | 0.13 | 0.0 | 0.50 |
| | | CHATGPT | 0.72 | **0.31** | 0.14 | **0.10** | **0.75** |
| MISCELLANEOUS | Adult-census | TAPAS-WTQ | 0.59 | 0.12 | 0.34 | 0.06 | 0.25 |
| | | TAPEX-WTQ | 0.59 | 0.12 | 0.16 | 0.0 | 0.45 |
| | | OMNITAB | 0.04 | 0.01 | 0.13 | 0.01 | 0.50 |
| | | CHATGPT | **0.71** | **0.68** | **0.47** | **0.35** | **0.58** |
| | Mushrooms | TAPAS-WTQ | 0.68 | 0.12 | 0.27 | 0.03 | 0.33 |
| | | TAPEX-WTQ | 0.44 | 0.10 | 0.33 | 0.02 | 0.43 |
| | | OMNITAB | 0.17 | 0.03 | 0.13 | 0.02 | 0.50 |
| | | CHATGPT | **0.77** | **0.75** | **0.49** | **0.37** | **0.56** |

Table 8: Results for QA-TRL for the different proprietary tables.

# D  Additional results for Semantic Parsing

Table 10 compares the performance of the SP models on proprietary data. Based on the achieved results, RESDSQL performed worst in the MEDICINE category. To delve deeper into the results, we focused on the table displaying the lower metric score, which can be found in Table 11. `Heart-attack` exhibits remarkable low values for cell recall across all `where` SQL categories. These inaccuracies can be attributed to a misprojection made by the model during prediction. The presence of the *output* column in the `heart-attack` table appears to confuse the model when it attempts to project all columns. our hypothesis is confirmed by the presence of the same error in the SELECT-ALL category.

| Table category | SQL category | Cell precision | Cell recall | Tuple cardinality | Tuple constraint | Tuple order |
|---|---|---|---|---|---|---|
| | SELECT-ALL | 0.00 | 0.00 | 0.00 | 0.00 | |
| | SELECT-ADD-COL | 0.43 | 0.03 | 0.03 | 0.03 | |
| | SELECT-RANDOM-COL | 0.38 | 0.07 | 0.02 | 0.02 | |
| | ORDERBY-SINGLE | 0.00 | 0.00 | 0.00 | 0.00 | 0.00 |
| | DISTINCT-MULT | 0.40 | 0.10 | 0.01 | 0.01 | |
| Sales-transactions | DISTINCT-SINGLE | 1.00 | 0.28 | 0.28 | 0.28 | |
| | WHERE-CAT-MAX-VALUES | 0.10 | 0.03 | 0.20 | 0.00 | |
| | WHERE-CAT-MIN-VALUES | 0.05 | 0.01 | 0.10 | 0.00 | |
| | WHERE-NUM-MAX-VALUES | 0.00 | 0.00 | 0.00 | 0.00 | |
| | WHERE-NUM-MEAN-VALUES | 0.00 | 0.00 | 0.00 | 0.00 | |
| | WHERE-NUM-MIN-VALUES | 0.00 | 0.00 | 0.00 | 0.00 | |
| | SELECT-ALL | 0.00 | 0.00 | 0.00 | 0.00 | |
| | SELECT-ADD-COL | 0.33 | 0.04 | 0.03 | 0.03 | |
| | SELECT-RANDOM-COL | 0.30 | 0.12 | 0.04 | 0.03 | |
| | ORDERBY-SINGLE | 0.00 | 0.00 | 0.00 | 0.00 | 0.00 |
| | DISTINCT-MULT | 0.33 | 0.18 | 0.18 | 0.18 | |
| Late-payment | DISTINCT-SINGLE | 0.97 | 0.45 | 0.46 | 0.45 | |
| | WHERE-CAT-MAX-VALUES | 0.08 | 0.02 | 0.01 | 0.00 | |
| | WHERE-CAT-MIN-VALUES | 0.08 | 0.02 | 0.01 | 0.00 | |
| | WHERE-NUM-MAX-VALUES | 0.00 | 0.00 | 0.00 | 0.00 | |
| | WHERE-NUM-MEAN-VALUES | 0.00 | 0.00 | 0.00 | 0.00 | |
| | WHERE-NUM-MIN-VALUES | 0.01 | 0.00 | 0.01 | 0.00 | |

Table 9: Results for QA chatGPT on Sales-transactions and Late-payment.

| Table category | Table name | Model | Cell precision | Cell recall | Tuple cardinality | Tuple constraint | Tuple order |
|---|---|---|---|---|---|---|---|
| ECOMMERCE | Sales-transactions | RESDSQL | 0.91 | 0.90 | 0.90 | 0.83 | 1.00 |
| | | GAP | 0.85 | 0.80 | 0.82 | 0.71 | 0.94 |
| | | UNIFIEDSKG | 0.79 | 0.77 | 0.76 | 0.74 | **1.00** |
| | | CHATGPT | **1.00** | **1.00** | **0.99** | **1.00** | **1.00** |
| | Fitness-trackers | RESDSQL | 0.90 | 0.88 | 0.94 | 0.79 | **1.00** |
| | | GAP | 0.84 | 0.80 | 0.80 | 0.74 | **1.00** |
| | | UNIFIEDSKG | 0.65 | 0.65 | 0.65 | 0.65 | **1.00** |
| | | CHATGPT | **1.00** | **1.00** | **1.00** | 0.90 | **1.00** |
| FINANCE | Account-fraud | RESDSQL | 0.89 | 0.86 | 0.95 | 0.77 | **1.00** |
| | | GAP | 0.79 | 0.77 | 0.75 | 0.73 | **1.00** |
| | | UNIFIEDSKG | 0.80 | 0.76 | 0.75 | 0.68 | **1.00** |
| | | CHATGPT | **0.99** | **0.99** | **0.98** | **0.90** | **1.00** |
| | Late-payment | RESDSQL | 0.90 | 0.87 | 0.95 | 0.77 | **1.00** |
| | | GAP | 0.80 | 0.79 | 0.78 | 0.74 | **1.00** |
| | | UNIFIEDSKG | 0.77 | 0.76 | 0.71 | 0.65 | 0.95 |
| | | CHATGPT | **1.00** | **1.00** | **1.00** | **0.90** | **1.00** |
| MEDICINE | Heart-attack | RESDSQL | 0.88 | 0.67 | 0.93 | 0.57 | 0.90 |
| | | GAP | **0.92** | 0.85 | 0.88 | 0.79 | 0.93 |
| | | UNIFIEDSKG | 0.72 | 0.66 | 0.71 | 0.61 | 0.90 |
| | | CHATGPT | **0.92** | **0.92** | **0.99** | **0.92** | **1.00** |
| | Breast-cancer | RESDSQL | 0.84 | 0.83 | 0.95 | 0.77 | **1.00** |
| | | GAP | 0.64 | 0.63 | 0.60 | 0.56 | 0.25 |
| | | UNIFIEDSKG | 0.72 | 0.71 | 0.70 | 0.70 | **1.00** |
| | | CHATGPT | **0.95** | **0.95** | **0.99** | **0.95** | **1.00** |
| MISCELLANEOUS | Adult-census | RESDSQL | 0.92 | 0.88 | 0.90 | 0.76 | **1.00** |
| | | GAP | 0.71 | 0.67 | 0.58 | 0.60 | **1.00** |
| | | UNIFIEDSKG | 0.69 | 0.64 | 0.63 | 0.58 | 0.95 |
| | | CHATGPT | **0.99** | **0.99** | **0.98** | **0.83** | **1.00** |
| | Mushrooms | RESDSQL | 0.96 | 0.91 | 0.91 | 0.78 | **1.00** |
| | | GAP | 0.94 | 0.89 | 0.89 | 0.80 | **1.00** |
| | | UNIFIEDSKG | 0.80 | 0.74 | 0.72 | 0.61 | **1.00** |
| | | CHATGPT | **0.99** | **0.99** | **0.99** | **0.99** | **1.00** |

Table 10: Results for SP-TRL for the different proprietary tables.

| SQL category | Cell precision | Cell recall | Tuple cardinality | Tuple constraint | Tuple order |
|---|---|---|---|---|---|
| SELECT-ALL | 0.00 | 0.00 | 0.00 | 0.00 | |
| SELECT-ADD-COL | 1.00 | 0.79 | 1.00 | 0.56 | |
| SELECT-RANDOM-COL | 1.00 | 0.72 | 1.00 | 0.40 | |
| ORDERBY-SINGLE | 1.00 | 0.99 | 0.91 | 0.91 | 1.00 |
| DISTINCT-SINGLE | 1.00 | 1.00 | 1.00 | 1.00 | |
| DISTINCT-MULT | 1.00 | 0.79 | 0.62 | 0.51 | |
| WHERE-CAT-MAX-VALUES | 1.00 | 0.28 | 1.00 | 0.20 | |
| WHERE-CAT-MIN-VALUES | 1.00 | 0.28 | 1.00 | 0.20 | |
| WHERE-NUM-MAX-VALUES | 1.00 | 0.86 | 0.98 | 0.83 | |
| WHERE-NUM-MEAN-VALUES | 1.00 | 0.78 | 1.00 | 0.75 | |
| WHERE-NUM-MIN-VALUES | 0.80 | 0.71 | 0.80 | 0.70 | |

Table 11: RESDSQL results for the `heart-attack` table.

