# OpenReview forum: "QATCH: Benchmarking SQL-centric tasks with Table Representation Learning Models on Your Data"
_NeurIPS.cc/2023/Track/Datasets_and_Benchmarks — NeurIPS 2023 Datasets and Benchmarks Poster_

### Official Review · Reviewer_dxvp · 2023-07-20
**Meets all the expectations for a Datasets & Benchmark Track paper**

**Rating:** 7
**Confidence:** 4
**Clarity:** The paper overall is very well written.

**Strengths:**

- The benchmarking approach is well motivated
- Some key insights are provided that highlight a number of interesting future directions
- Open source and well documented code base

**Additional Feedback:**

Could you produce a large and diverse benchmark (using say 100s of tables across 10s of domains) and share the benchmark as a dataset, similar to Spider and other “rigid” benchmarks? I believe there is a tremendous value in doing so.

**Correctness:**

Most claims seem to be correct and reasonable. The small number of tables used in the experiments may result in reporting outlier or uncommon conditions.

**Documentation:**

I haven’t tried to reproduce the benchmark, but the code seems to be in a reasonable shape. The most disappointing aspect is the non-commercial license, which limits its potential for real impact, which is the main motivation behind this work.

**Ethics:**

I do not see any ethical concerns.

**Limitations:**

Some limitations are discussed under discussions. Limitation of the approach for tasks other than QA and SP are not discussed.

**Opportunities For Improvement:**

- Code is shared under a non-commercial license, limiting its potential real-world impact
- The number of datasets (8) seems small. Given that the approach seems very efficient, couldn’t you include more (possibly many more) tables?
- Some tasks, such as Table Metadata Prediction (TMP) may not benefit from this approach as it’s not clear how one can generate the ground truth

**Relation To Prior Work:**

Related work is discussed well.

The choice of TRL models is not explained well. For example, could TABERT have been used instead or in addition?

**Summary And Contributions:**

The paper presents a mechanism of generating a custom benchmark for evaluation of the performance of table representation learning (TRL) based methods for two tasks. The work is well motivated, the solution is described in detail, and experimental evaluation over 8 custom datasets provide some interesting insights on the performance of different TRL based methods. There is a good discussion of lessons learned and areas for future work.

---

> ### Author Response · Authors · 2023-08-18
>
> **General comment**: We would like to thank the reviewer for her/his insightful comments and suggestions, which allow us to improve the quality of our work. We will change the license of our project code, clarify better the rationale behind our work and the tasks for which our tool is suited to. Finally, we will comment on the applicability to other tasks such Table Metadata Prediction.
>
> **Reviewer’s concern**: Licence of the project code.\
> **Authors’ answer**: We will follow the reviewer’s suggestion and change the license to enable commercial use.
>
> **Reviewer’s concern**: Request for a large number of datasets.\
> **Authors’ answer**: While QATCH can automatically produce a benchmark for 100s of tables, this is not our goal. Just with 8 tables, we already see a clear distinction in the results between those obtained from proprietary datasets and from Spider, even across various proprietary tables. While we are open to adding more tables as suggested by the reviewers, our goal is not to create a new corpus, as our argument is that tests should be created ad-hoc on the proprietary datasets.
>
> We show that different models have the best performance over different datasets and task pairs. As every organization has proprietary tables, the evaluation should be done locally, by the users, without relying on a corpus with questions and datasets that can differ greatly from the proprietary use case. Our QATCH tool enables such evaluation in an automatic fashion by using queries of increasing complexity.
> The tool is publicly available, through Google Colab, at the following link:
> https://colab.research.google.com/drive/1SNoy3GZGPWltVS5cL068xAG9YoPS_3_l?usp=sharing#scrollTo=jErKRJe70PYi}.
>
> **Reviewer’s concern**: Applicability of the tool to tasks other than QA and SP.\
> **Authors’ answer**: We agree with the reviewer that some tasks, such as Table Metadata Prediction, may not benefit from this approach and we will change the title to *Benchmarking SQL-centric tasks with Table Representation Learning Models on Your Data* and revise the abstract and introduction accordingly. We will clarify in the limitation discussion that the proposed solution does not cover other TRL tasks.
>
> **Reviewer’s concern**: The choice of TRL models is not explained well. For example, could TABERT have been used instead or in addition?\
> **Authors’ answer**: We restricted the selection of the model to the SOTA and the available fine-tuned models. There are many models in the literature that are only available as encoders with little emphasis on the actual task, and unfortunately TaBERT is one of them. Indeed, TaBERT authors have never updated the semantic parsing reproducibility section ( https://github.com/pcyin/pytorch_neural_symbolic_machines/#training-with-pre-trained-tabert-models ). However, we are open to include any other model suggested by the reviewer.

---

> > ### Comment · Reviewer_dxvp · 2023-08-21
> >
> > Thank you for the response. It's great that you are flexible with the license, and I like your new title. I believe I also understand the issue with TaBERT, although it would be nice to mention this in the paper and perhaps also reach out to TaBERT authors to make sure your statement is accurate.
> >
> > I believe this is a good paper and should be accepted.

---

### Official Review · Reviewer_aVca · 2023-07-20
**A toolbox for validating the performance of TRL models on proprietary datasets**

**Rating:** 6
**Confidence:** 4

**Strengths:**

This manuscript presents the QATCH toolbox for automatic generation and evaluation of test check-lists customized for proprietary data, which enriches research on the specificity and adaptability of TRL models. The authors experimentally evaluate the performance of a number of TRL models in question-answering tasks.

**Additional Feedback:**

The paper should consider the soundness of its experimental design and enrich the experiment to support their innovative ideas. In addition, authors should explain the rationality of the cross-task performance metrics used.

**Clarity:**

While this paper contains some interesting ideas and concrete examples, the writing could be improved in some areas. For example, some parts may be too colloquial, and some concepts may not be emphasized enough. I suggest that the authors consider revising the paper to improve its clarity and readability.

**Correctness:**

This paper merely provides a toolbox for validating the performance of TRL models on proprietary datasets, but does not make any substantial contribution to the creation of the datasets. If this paper is considered a benchmark, it submits some experiments, but these are not sufficient. This paper uses a new set of cross-task performance metrics, but does not explain their rationale.

**Documentation:**

In terms of datasets, the experiments in this paper are conducted on proprietary Kaggle without any substantial contribution to the creation of the dataset, and the paper does not provide URLs for accessing this dataset. In terms of benchmarks, the paper provides detailed experiments on two tasks, but the results are not convincing.

**Ethics:**

no.

**Limitations:**

While the main contribution of this paper is to validate the performance of the TRL model on a proprietary dataset, partial experiments on a benchmark dataset should also be conducted to demonstrate the validity of the methodology and model. The lack of comparison with the SOTA makes the experiments less credible. We appreciate the motivation of this paper, and it is a good work to evaluate the performance of TRL models on proprietary datasets, but based on the experiments in this paper alone, we still do not know which TRL model is advantageous on which proprietary dataset.

**Opportunities For Improvement:**

1.	The main work of this paper is to validate the performance of TRL models on proprietary datasets, but does not contribute substantially to the creation of the dataset. Therefore, the novelty and innovation of the work in this paper is slightly lacking.
2.	This paper proposes QATCH-Generate, which generates SQL declaration, its free-text version, and the expected ground truth consisting of table instances, but it is not clear how to ensure that each of the generated objects is correct and one-to-one.
3.	A set of cross-task performance metrics are used in this paper. While examples of each metric are given, the strengths and weaknesses of each metric, as well as the tasks and scenarios to which each metric applies, are not provided. For example, measuring the order of tuple outputs is mentioned in Section 2, but it is not clear why the order of tuple outputs needs to be measured and which tasks require that the tuple outputs are in the correct order.
4.	In QA task experiments in Table 3, the results and analysis are not convincing. The focus of this paper is to evaluate the performance of TRL models on each of the proprietary datasets, but none of the TRL models perform well on any of the datasets, which casts doubt on the work of this paper. Furthermore, the authors do not explain why the TAPEX model performs poorly on almost all datasets. Experiments on the benchmark dataset Spider also use the metrics and TRL models presented in the paper, but it is not clear what the authors are trying to show in this column.
5.	In the case of the SP task experiments (Table 4), while there are related experiments in the Supplementary Materials, there is very little in the body of the text about the SP task.

**Relation To Prior Work:**

This paper does not cite several important papers on this topic, nor does it compare with published SOTA methods.

**Summary And Contributions:**

This paper describes a toolbox called QATCH for testing the performance of representation learning (TRL) models on proprietary datasets. The toolbox helps organizations validate the performance of TRL models on their proprietary datasets. The authors validate the performance of three TRL models on QA tasks and one TRL model on SP tasks on 8 Kaggle proprietary datasets. In addition, the authors evaluate the performance of TRL models on a number of cross-task performance metrics.

---

> ### Author Response · Authors · 2023-08-18
>
> ***OFFICIAL COMMENT PART 1***
>
> **General comment**: We would like to thank the reviewer for her/his insightful comments and suggestions, which allow us to improve the quality of our work. We understand that we need to better clarify the scope of our research work and the formulation and use of the SQL declarations. We will also improve the presentation of the results and the explanation of the proposed metrics.
>
> **Reviewer’s concern**: Scope of the research and pertinence to the Datasets and Benchmarks NEURIPS'23 track.\
> **Authors’ answer**: According to the CFP for the *Dataset and Benchmark* track, the proposal of *benchmarking tools for evaluation* is welcome and the scope for contribution is not restricted to the creation of new datasets.\
> Quoted from the official list: *"Benchmarks on new or existing datasets, as well as benchmarking tools"* \
> (https://neurips.cc/Conferences/2023/CallForDatasetsBenchmarks ).\
> The novelty of our proposal is in the release and evaluation of a new testing benchmark suited to evaluate TRL models on proprietary data.\
> To the best of our knowledge, our work is the first testing benchmark to evaluate TRL models on proprietary data.
>
> **Reviewer’s concern**: Correctness and uniqueness of the SQL outputs.\
> **Authors’ answer**: The three objects (SQL declaration, its free-text version, and the expected ground truth) are correct by construction.
> - The SQL declaration is constrained by the input schema and the SQL grammar.
> - The pair (SQL declaration, free text question) are fixed by manually written templates (Table 1) to avoid hallucinations. The natural language used in the question is intentionally simple and not ambiguous to obtain a one-to-one mapping.
> - The ground truth is obtained by executing the SQL with a DB engine over the initial table. Even if the SQL script had a mistake, it would fail during this execution.
>
> **Reviewer’s concern**: Rationale behind the proposed metrics.\
> **Authors’ answer**: We will  better clarify the different aspects captured by the metrics by adding more discussion and examples showing the complementary of the proposed metrics. For example, consider the expected output for a query with two tuples (a1,b1) (a2,b2) and one model returning as output (a1,b2) (a2,b1). In this case, only the *tuple constraint* metric captures the issue, i.e., precision, recall and cardinality would give perfect scores.
>
> We plan to add to the paper the following strengths and weaknesses of each metric:
> - *Cell precision*: the higher is the score, the more predicted elements are in the target. However, it does not measure how many target cells are in the predictions (measured by *cell recall*).
> - *Cell recall*: the higher is the score, the more target cells are in the prediction. It does not measure how many prediction cells are in the target (measured by *cell precision*).
> - *Tuple constraint*: it is one if the expected and produced outputs have both the same schema, the same cardinality and the same cell values, zero otherwise. However this hard constraint does not capture all the cases (see answer for the first Reviewer for *tuple cardinality*).
> - *Tuple cardinality*: It is a "softer" constraint w.r.t. the *tuple constraint* since it does not consider neither the schema nor the cell values. However, it has to be analysed with *cell precision* and *cell recall* to be meaningful.
>  - *Tuple order*: it is the only metric to check whether the prediction has the required order or not.
>
> Regarding the *tuple order*, we state that such a measure is applied only for SQL queries that contain the ORDER BY clause (line 232 in the submission).
>
> **Reviewer’s concern**: In-depth analysis of QA results.\
> **Authors’ answer**: One of the insights of our work is that existing models do relatively well on SPIDER, while they have more scattered results over proprietary datasets (lines 290-292 in the submission). Our goal is to show that the scenarios in SPIDER do not reflect the reality of enterprise schema. Specifically, tabular benchmarks like SPIDER proved to be unreliable to evaluate TRL model effectiveness in enterprise scenarios. For example, for all models the *cell recall* evaluated with proprietary table is lower (for TAPAS and TAPEX much lower) than the same model evaluated with SPIDER (lines 271-273).
>
> As we discuss at the end of Sec. 4 in the submission, the poor results are explained by multiple factors, including the roles played by the dataset used in the fine tuning (WTQ for TAPAS and TAPEX, which lacks complex questions) and by the more challenging domain (and vocabulary) of the proprietary tables (most Spider tables contain very popular attributes that have been seen by the TRL models at pre-training). In the supplementary materials (lines 53-59 and Table 1) there is a more in depth analysis of the difference between SPIDER and the selected proprietary tables. We will revise such discussion to add more examples to make these points clearer.

---

> > ### Author Response · Authors · 2023-08-18
> >
> > ***OFFICIAL COMMENT PART 2***
> >
> > **Reviewer’s concern**: Presentation of SP results.\
> > **Authors’ answer**: We decided to focus our study on the QA task as it is more challenging than the SP one. However, we agree with the reviewer that we could move Table 6 to the main body and extend its discussion thanks to the one additional page allowed for accepted papers.
> >
> > **Reviewer’s concern**: Scope of the research work.\
> > **Authors’ answer**: We have reported results for available fine-tuned SOTA models, but we are happy to include any public fine-tuned model that the reviewer would like to see included in the comparison. Unfortunately, several open models are just available as encoders (with no fine-tuned version).
> >
> > We remark that our goal is not to propose a new model or to create a ranking of those available. As reported in Table 6, there is no model that excels on every proprietary dataset for every metric. As there is no winning model in all settings, our tool provides a pragmatic automatic solution to the challenging problem of selecting the right model for the given dataset and task at hand.
> >
> > **Reviewer’s concern**: Comparison with prior work.\
> > **Authors’ answer**: We neither try to create a large corpus of new datasets nor propose a new version of SPIDER. Similarly, we are not trying to argue that there is a winning method that should always be used in a specific setting. Conversely, we show that different models achieve the best performance over different datasets and task pairs. As every organization has proprietary tables, the evaluation should be done locally, by the users. Our tool allows end-users to automate such an evaluation with performance metrics that capture different quality aspects. The results also show that proprietary data are, in general, quite different from benchmark data (e.g., SPIDER).\
> > We will include and discuss any important paper on this topic that the reviewer wants to bring to our attention.
> >
> > **Reviewer’s concern**: Lack of reproducibility.\
> > **Authors’ answer**: We have released the URL to access the datasets in the GitHub repository (https://github.com/spapicchio/QATCH#install-and-prepare-data ).
> > To improve content accessibility and simplify experiments' reproducibility, we will include such URL also in the main paper body.

---

> ### Comment · Reviewer_aVca · 2023-08-22
>
> The author partially addressed my concerns. However, based on the current state of the paper and the author's response, I stand by my original score.

---

> > ### Author Response · Authors · 2023-08-22
> >
> > We kindly ask the reviewer to tell us what concerns are not addressed.
> > 1. Our work is in scope with the CFP.
> > 2. We guarantee correctness of the tests.
> > 3. We clarified the differences in the metrics.
> > 4. It is our goal to show that models fail on proprietary data (this is one of the core results of the work).
> > 5. We agree that we will move content from the appendix to the main body.

---

> > ### Comment · Reviewer_aVca · 2023-08-28
> >
> > The authors' latest response and additional experimental resutls convinced me to change my score.

---

### Official Review · Reviewer_U5Jd · 2023-07-21
**QATCH: Benchmarking Table Representation Learning Models on Your Data**

**Rating:** 6
**Confidence:** 4
**Correctness:** Yes.
**Clarity:** Yes.

**Strengths:**

1. A benchmark of table representation learning is proposed.

2. An evaluation framework is proposed, avoiding trusting the results achieved on the existing benchmarks across different tasks, as they are not always replicable on proprietary data.

3. The experiments reveal some interesting findings.

**Additional Feedback:**

1. The title of the paper is "Benchmarking Table Representation Learning Models on Your Data". However, only a small subset of tasks are covered in the paper. For example, in "TURL: Table understanding through representation learning", entity linking, column type annotation, relation extract, etc., were also considered. It is suggested to consider more tasks, or modify the title by qualifying the scope to QA and SP.

2. It is suggested to evaluate more TRL methods. In addition, many of them are named in the related work part (lines 98-102, page 3), but lacking description. A more detailed discussion would help understand the differences of these TRL methods.

3. Only RESDSQL and ChatGPT are evaluated for SP. Why are TAPAS and TAPEX not available here?

**Documentation:**

Yes.

**Ethics:**

There are no ethical concerns.

**Limitations:**

I don't see limitations in terms of potential negative societal impact.

**Opportunities For Improvement:**

1. The paper only covers question answering and semantic parsing, which are only a subset of the tasks for which table representations are trained.

2. The proposed framework works on SQL, while many TRL tasks are not handled in an RDBMS (e.g., web table retrieval).

3. TRL has been extensively studied in the last few years. However, only four models are evaluated on this benchmark. Moreover, one of them is ChatGPT, which is not specific to TRL, though available for solving QA and SP.

Request for revision (after rebuttal):

1. Please focus on SQL-centric tasks and modify title, abstract, and introductions accordingly.

2. Please report the experimental settings, results, and discussions during the rebuttal period in the paper.

**Relation To Prior Work:**

Yes.

**Summary And Contributions:**

This paper presents QATCH, a testing benchmark for table representation learning models. It provides end-users with a flexible and adaptive solution for the automatic assessment of models' performance on proprietary data. Experiments are performed to evaluate question answering and semantic parsing. Results show that existing questions in benchmarks do not capture important properties of custom tables, and popular metrics fail short in measuring the quality of the models' output.

---

> ### Author Response · Authors · 2023-08-18
>
> **General comment**: We would like to thank the reviewer for her/his insightful comments and suggestions, which allow us to improve the quality of our work. We will better clarify the motivations behind our TRL model selection and the rationale behind our work.
>
> **Reviewer’s concern**: Better clarify the scope of the research.\
> **Authors’ answer**: Following the reviewer's suggestion, we will change the title to *Benchmarking SQL-centric tasks with Table Representation Learning Models on Your Data* and revise the abstract and introduction accordingly. We will also clarify in the limitation discussion that the proposed solution does not cover other TRL tasks such as web table retrieval.
>
> **Reviewer’s concern**: Motivate the TRL model selection, along with the use of ChatGPT.\
> **Authors’ answer**: Our TRL model selection has been restricted to those fine tuned from existing research papers or established open source repositories.  According to our empirical findings, there is no model that excels on every proprietary dataset for every metric.
> We will include results for pre-trained TRL models that the reviewer can suggest for the two tasks considered in our paper.
>
> Our main goal is to show the role of proprietary datasets and complex queries (questions), rather than ranking a large array of models.
> Our tool provides a pragmatic automatic solution to the challenging problem of selecting the right TRL model for the given proprietary dataset and task at hand. We will better clarify this point in the main body of the paper.
>
> Concerning ChatGPT, we will clarify that is not a TRL model, but we plan to keep it as an interesting reference for results that can be obtained with state of the art, proprietary neural approaches for QA and SP.
>
> **Reviewer’s concern**: The results for SP achieved by TAPAS and TAPEX are not available.\
> **Authors’ answer**: TAPAS and TAPEX are not competitive for this task - as stated in TAPEX’s original paper: *We have tried to apply TAPEX for a text-to-SQL task, where the input remains the same and the output converts to SQL. However, TAPEX does not show a significant advantage over BART*. We will better clarify this point in the paper.
>
> **Reviewer’s concern**: Add a more detailed description of the TRL methods.\
> **Authors’ answer**: We agree with the reviewer
> that the first paragraph of the related work part (lines 98-102 in the current submission)
> is quite generic and not focused on the QA and SP tasks under analysis.
> Since our main goal is to design a testing tool for SQL-centric tasks with Table Representation Learning Models,
> we will shorten the introductory paragraph and extend the description of the SQL-centric tasks, i.e., Question Answering and Semantic Parsing for TRL models. We will provide a more detailed description of the corresponding models (e.g., TAPAS, TAPEX, RESDSQL).

---

> > ### Comment · Reviewer_U5Jd · 2023-08-21
> > **Re: Official Comment by Authors**
> >
> > The authors partially addressed my concerns. However, based on the current form of the paper and the authors' response, I would keep my rating. In general, the scope of this paper should be qualified to the tasks and models evaluated. Also, the evaluation should be made more thorough. I think considerable efforts are needed to revise the paper and address the drawbacks.

---

> > > ### Comment · Reviewer_dxvp · 2023-08-21
> > >
> > > I am a reviewer in favor of accepting this paper. May I ask why you believe "the evaluation should be made more thorough"? This paper presents a benchmarking tool, and the paper proves that the tool is useful. The results and the lessons learned are already quite useful. They do not claim a thorough evaluation of all TRL models, so why should they perform such an evaluation? I would agree with the assessment if this was not a datasets and benchmarking track paper.

---

> > > > ### Comment · Reviewer_U5Jd · 2023-08-21
> > > > **Response to Reviewer dxvp**
> > > >
> > > > First, the title of the paper is "QATCH: Benchmarking Table Representation Learning Models on Your Data". Table representation learning is a very popular topic, and various tasks have been targeted in the last decade (entity linking, column type annotation, relation extraction, etc.). In this paper, only two tasks, QA and SP, are evaluated. Therefore, I don't think the title and the claim in the abstract ("Our purpose is to ... highlight TRL models’ strengths and weaknesses on relational tables unseen at training time) are appropriate.
> > > >
> > > > Second, even narrowing down the scope to QA and SP, various methods have been proposed. For QA, please see Table 3 in a survey paper [1]. For SP, please see Appendix A.3 in [1] and [2, 3]. I'm not suggesting the authors consider all of these methods, but the current evaluation of this paper, which considers only two TRL methods for QA (ChatGPT is NOT a TRL method) and only one method for SP, is overshadowed by those in existing dataset and benchmarking papers on this topic [2, 4]. Note that [2, 4] target a much smaller scope, which is subsumed by the tasks that can be handled by TRL. This means, this paper, is trying to support its overclaimed title, abstract, and the contributions in the introduction, with a less convincing evaluation. For example, for SP, there is no comparison with other methods, and thus we don't even know if the dataset is hard enough for existing TRL methods.
> > > >
> > > > [1] Table Pre-training: A Survey on Model Architectures, Pretraining Objectives, and Downstream Tasks. IJCAI 2022.
> > > >
> > > > [2] Spider: A Large-Scale Human-Labeled Dataset for Complex and Cross-Domain Semantic Parsing and Text-to-SQL Task. EMNLP 2018.
> > > >
> > > > [3] RESDSQL: Decoupling Schema Linking and Skeleton Parsing for Text-to-SQL. AAAI 2023.
> > > >
> > > > [4] TabFact: A Large-scale Dataset for Table-based Fact Verification. ICLR 2020.
> > > >
> > > > Based on the above argument, my conclusion is that the paper is below the acceptance threshold, and I would keep my score.

---

> > > > > ### Author Response · Authors · 2023-08-25
> > > > >
> > > > > We believe there is a misunderstanding with reviewer U5Jd.
> > > > >
> > > > > We already ran the models that are suggested in the comment above that satisfy our requirements, i.e,
> > > > > 1. They are tailored for the task (either QA or SP).
> > > > > 2. They are available online.
> > > > >
> > > > > RESDSQL [2] satisfies both, and indeed it is the method reported in our submission for SP.
> > > > >
> > > > > Other models are not included for the sake of space as they do not add significant material w.r.t. our main contribution. Let us state again that we are not after a benchmarking of the available models, our goal is to show that models should be tested on proprietary datasets.
> > > > >
> > > > > To clarify our point, please consider one SP model cited in the suggested survey [1] (Table 3): "UnifiedSKG: Unifying and Multi-Tasking Structured Knowledge Grounding with Text-to-Text Language Models" (https://arxiv.org/abs/2201.05966).
> > > > >
> > > > > We obtain the following results:
> > > > >
> > > > > | Db Id         | Cell Precision Skg | Cell Recall Skg | Tuple Cardinality Skg | Tuple Constraint Skg | Tuple Order Skg |
> > > > > |---------------|--------------------|-----------------|-----------------------|----------------------|-----------------|
> > > > > | ecommerce     | 0.71               | 0.71            | 0.69                  | 0.69                 | 1.00           |
> > > > > | finance       | **0.79**               | 0.76            | 0.74                  | 0.67                 | 0.98            |
> > > > > | medicine      | 0.72               | 0.69            | 0.70                  | 0.66                 | 0.95            |
> > > > > | miscellaneous | 0.74               | 0.69            | 0.68                  | 0.59                 | 0.98            |
> > > > > | SPIDER DEV    | 0.77               | **0.79**            | **0.77**                  | **0.77**                 | **1.00**            |
> > > > >
> > > > > We can see that for most metrics the datasets from Spider get the best results. This confirmed the evidence from RESDSQL, which we report in the main body as it is has the best results. However, we can add the results above for UnifiedSKG to Table 6 and move it to the main body.
> > > > >
> > > > > Similarly, we have also tested the QA model  “OmniTab: Pretraining with Natural and Synthetic Data for Few-shot Table-based Question Answering” (https://arxiv.org/abs/2207.03637). The model has worse  performance w.r.t TAPAS and TAPEX and again confirms the claim of the paper i.e. the SPIDER results are higher than the proprietary ones.
> > > > >
> > > > > | Db Id         | Cell Precision OmniTab | Cell Recall OmniTab  | Tuple Cardinality OmniTab  | Tuple Constraint OmniTab  | Tuple Order OmniTab |
> > > > > |---------------|----------------|-------------|-------------------|------------------|-------------|
> > > > > | ecommerce     | 0.31           | 0.10        | 0.26              | 0.09             | 0.50         |
> > > > > | finance       | **0.36**           | 0.08        | 0.24              | 0.06             | 0.50         |
> > > > > | medicine      | 0.34           | 0.10         | 0.29              | 0.08             | 0.50         |
> > > > > | miscellaneous | 0.19           | 0.08        | 0.24              | 0.08             | 0.50         |
> > > > > | SPIDER        | 0.2            | **0.15**        | **0.37**              | **0.14**             | **0.52**        |
> > > > >
> > > > > We believe that TAPAS and TAPEX with their better performance already show that tests should be created ad-hoc on the proprietary datasets, but we can increase the number of models reported in the experiments by adding OmniTab.
> > > > >
> > > > > In general, the reviewer can point us to any fine tuned model for QA or SP and we can post the result on OpenReview in less than a day, plus include them in the paper.
> > > > >
> > > > > [1] Table Pre-training: A Survey on Model Architectures, Pretraining Objectives, and Downstream Tasks. IJCAI 2022.
> > > > >
> > > > > [2] RESDSQL: Decoupling Schema Linking and Skeleton Parsing for Text-to-SQL. AAAI 2023.

---

> > > > > > ### Comment · Reviewer_U5Jd · 2023-08-25
> > > > > > **Re: Official Comment by Authors**
> > > > > >
> > > > > > Thanks for the effort made by the authors.
> > > > > >
> > > > > > It is interesting to see the above results. I also suggest evaluating Grappa [1] and GAP [2] for SP, as they are models targeting text-to-SQL tasks and publicly available.
> > > > > >
> > > > > > [1] Grappa: Grammar-augmented pre-training for table semantic parsing. ICLR 2021. https://github.com/taoyds/grappa
> > > > > >
> > > > > > [2] Learning Contextual Representations for Semantic Parsing with Generation-Augmented Pre-Training. AAAI 2021. https://github.com/awslabs/gap-text2sql
> > > > > >
> > > > > > Since this work proposes a benchmark for evaluating TRL methods on QA and SP, I believe that the comparison of different TRL methods is important. This is also important for your claim that models should be tested on proprietary datasets, because such a conclusion should be generalized to various TRL models instead of only two (for QA) or even one (for SP). In addition, Table 6 shows that ChatGPT performs worse on Spider than on the proposed benchmark. Though ChatGPT is not a TRL model, I doubt if the observation that "for most metrics the datasets from Spider get the best results" can be generalized to other TRL models.

---

> > > > > > > ### Comment · Reviewer_U5Jd · 2023-08-25
> > > > > > > **Re: Official Comment by Authors**
> > > > > > >
> > > > > > > Following the above comments, I appreciate that the authors have been trying their best to address the concerns pointed in the review. However, as more experimental results are provided, I am moving towards rejection rather than acceptance of the paper. I am more convinced that the paper is below the acceptance level for the following reasons.
> > > > > > >
> > > > > > > 1. According to the guidelines of rebuttal, "original submission will serve as the basis for the reviewers' (and ACs') acceptance recommendations. The rebuttals should serve only to clarify the reviewers' and ACs' questions during the discussion period" (see https://nips.cc/Conferences/2023/PaperInformation/NeurIPS-FAQ). The experimental results provided in the rebuttal should have been reported in the original submission. In addition, the paper needs revision for its title, abstract, introduction, and conclusion, which requires heavy rewriting of the paper, to make the scope within SQL-centric tasks.
> > > > > > >
> > > > > > > 2. The original submission evaluates only 3 models for QA tasks and 2 models for SP tasks, even if we count ChatGPT as a TRL model. Variants of models are not considered, and there is no description of the setup of these models. Let's see the evaluation in existing dataset and benchmark papers on this topic: In "Spider: A Large-Scale Human-Labeled Dataset for Complex and Cross-Domain Semantic Parsing and Text-to-SQL Task", 5 models were compared. In "TabFact: A Large-scale Dataset for Table-based Fact Verification", 4 models were compared, with many variants (Table 2) considered. From the competitor perspective, the evaluation in this submission is not on a par with existing works in this category. Since TRL has received considerable attention in recent years, we are supposed to try more models before making any conclusions on the evaluation of TRL.
> > > > > > >
> > > > > > > 3. Following the above point, in this submission, the insights (proprietary data v.s. Spider) for TRL models were observed on very few TRL models (2 for QA and 1 for SP). Therefore, I doubt if such insights can translate to other TRL models. For example, for SP, we even don't know which T5 model (base, large, 3B) was chosen for RESDSQL, and if the same observations can be made for other T5 settings, let alone other language models. In the rebuttal, the authors provided the results of UnifiedSKG. However, UnifiedSKG also employs T5 as its language model. So, the claim "for most metrics the datasets from Spider get the best results" in the rebuttal could be attributed to the use of T5. The authors are encouraged to investigate this.
> > > > > > >
> > > > > > > 4. The role of ChatGPT is intriguing. Many TRL models serialize table contents and feed them to a language model. ChatGPT is also a language model. Table 6 shows ChatGPT performs well (most metrics reported are 0.97 to 1) on proprietary data but less competitive on Spider. This suggests that larger models could be exceptionally good and achieve close to 1 metrics on proprietary data. Therefore, as noted in my previous comments, it is unclear whether the proposed benchmark is hard enough for TRL models, because we expect people would resort to larger language models (e.g., LLaMA) to deal with tabular data very soon.

---

> > > > > > > > ### Author Response · Authors · 2023-08-27
> > > > > > > >
> > > > > > > > We thank the reviewer for the insightful discussion that will certainly lead to a stronger paper.
> > > > > > > >
> > > > > > > > We would like to clarify our point of view on the 4 main points raised in the last comment. We realize that they won't change the reviewer's opinion, but we hope they could be useful for the discussion when taking the final decision for our submission.
> > > > > > > >
> > > > > > > > 1. Regarding the polishing of the final version, it is good practice that comments in the reviews lead to a better, revised version of the original submission. We argue that the extra experiments (suggested by the reviewer in the discussion) and the more focused scoping do not change the main contributions of the paper.
> > > > > > > >
> > > > > > > > 2. Our contribution is not an experimental evaluation of existing methods, but rather a framework to evaluate them on proprietary data automatically. Our original choice has been to focus the paper on a small sets of models for the sake of space - the results from additional models confirm the insights from the small set of models (see also 3 below).
> > > > > > > > However, given the remarks from the reviewer, we understand that more reported models can make the message stronger and are willing to add experimental results for all the methods discussed to the main body of the paper.
> > > > > > > >
> > > > > > > > 3. We found other models for SP to have much lower performance and, again, we are not trying to review alternative models in an exhaustive way. However, the argument about T5 is valid.  To understand whether the worst results on the proprietary datasets were caused by T5 or not, we have evaluated one of the models suggested in the previous comment which is not based on T5 [1]. The following results confirm the claim of the paper i.e. the SPIDER results are higher than the proprietary ones:
> > > > > > > >
> > > > > > > > | Db Id         | Cell Precision Gap | Cell Recall Gap | Tuple Cardinality Gap | Tuple Constraint Gap | Tuple Order Gap |
> > > > > > > > |---------------|--------------------|-----------------|-----------------------|----------------------|-----------------|
> > > > > > > > | ecommerce     | 0.84               | 0.80            | 0.81                  | 0.73                 | 0.97            |
> > > > > > > > | finance       | 0.79               | 0.78            | 0.76                  | 0.74                 | **1.00**            |
> > > > > > > > | medicine      | 0.77               | 0.73            | 0.73                  | 0.67                 | 0.59            |
> > > > > > > > | miscellaneous | 0.82               | 0.78            | 0.73                  | 0.69                 | **1.00**            |
> > > > > > > > | SPIDER DEV    | **0.98**               | **0.98**            | **0.95**                  | **0.93**                 | 0.96            |
> > > > > > > >
> > > > > > > > 4. We believe the last reviewer's comment confirms that reporting GPT's results in the paper is valuable as it enables a comparative analysis with TRL models with interesting insights. Please notice that results for GPT's are good for SP (suggesting that methods are mature for this task), but not for QA.
> > > > > > > >
> > > > > > > > [1] Learning Contextual Representations for Semantic Parsing with Generation-Augmented Pre-Training. AAAI 2021. https://github.com/awslabs/gap-text2sql

---

> > > > > > > > > ### Comment · Reviewer_U5Jd · 2023-08-28
> > > > > > > > >
> > > > > > > > > Thanks for providing more results. Along with the results on previous comments, they would significantly improve the experiment part of the paper.
> > > > > > > > >
> > > > > > > > > Although ChatGPT is non-TRL and used as a reference here, I believe its performance needs more investigation (e.g., by a comparison of win/loss examples on both datasets), because unlike the results of TRL methods given by the authors, the SP test of ChatGPT suggests otherwise. That is, it is a counterexample to the claim that proprietary data is harder than Spider, hence compromising its usefulness, especially when considering the fact that ChatGPT's performance is close to 1 on proprietary data but much worse on Spider. It could be due to its large model size, or possible use of proprietary data (or something with similar distributions) in RLHF. However, I don't find any discussion on this abnormal behavior. This is also one of the reasons why I insist on testing more models to justify the usefulness of the proposed benchmark.
> > > > > > > > >
> > > > > > > > > In addition, what is the T5 model setting (base, large, or 3B) for RESDSQL? Have you tried other model settings and confirmed that the gap between the proposed benchmark and Spider can be witnessed as well?

---

> > > > > > > > > > ### Author Response · Authors · 2023-08-28
> > > > > > > > > >
> > > > > > > > > > For ChatGPT, our explanation is that it does very well also on proprietary datasets as those are taken from the Web, so they are probably part of its training data. We can extend the evaluation by using datasets that are not available on the Web by crafting them (e.g., fake data about people). Another direction is to consider an open model, such as Llama 2, but this direction could steer the paper into the evaluation of LLMs, which is not the focus of this submission.
> > > > > > > > > >
> > > > > > > > > > Regarding the T5 model settings, we agree with the reviewer and we will add the specification LARGE in the main paper. In addition, we have also run RESDSQL with the missing T5 versions, i.e. base and 3B. Since SPIDER results are higher than the proprietary ones, the following results are in line with the claim of the paper:
> > > > > > > > > >
> > > > > > > > > > | Db Id         | Cell Precision Resdsql Base | Cell Recall Resdsql Base | Tuple Cardinality Resdsql Base | Tuple Constraint Resdsql Base | Tuple Order Resdsql Base |
> > > > > > > > > > |---------------|-----------------------------|--------------------------|--------------------------------|-------------------------------|--------------------------|
> > > > > > > > > > | ecommerce     | 0.84                        | 0.80                     | 0.80                           | 0.72                          | 0.61                     |
> > > > > > > > > > | finance       | 0.80                        | 0.75                     | 0.75                           | 0.64                          | 0.63                     |
> > > > > > > > > > | medicine      | 0.86                        | 0.83                     | 0.81                           | 0.71                          | 0.48                     |
> > > > > > > > > > | miscellaneous | 0.72                        | 0.67                     | 0.62                           | 0.50                          | 0.65                     |
> > > > > > > > > > | SPIDER DEV    | **0.95**                        | **0.95** |  **0.94**                           | **0.92**                          | **1.00**                     |
> > > > > > > > > >
> > > > > > > > > >
> > > > > > > > > > | Db Id         | Cell Precision Resdsql 3B | Cell Recall Resdsql 3B | Tuple Cardinality Resdsql 3B | Tuple Constraint Resdsql 3B | Tuple Order Resdsql 3B |
> > > > > > > > > > |---------------|---------------------------|------------------------|------------------------------|-----------------------------|------------------------|
> > > > > > > > > > | ecommerce     | 0.96                      | 0.93                   | 0.94                         | 0.86                        | **1.00**                   |
> > > > > > > > > > | finance       | 0.92                      | 0.90                   | 0.92                         | 0.82                        | **1.00**                   |
> > > > > > > > > > | medicine      | 0.94                      | 0.95                   | 0.94                         | 0.86                        | **1.00**                   |
> > > > > > > > > > | miscellaneous | 0.90                      | 0.88                   | 0.87                         | 0.75                        | **1.00**                   |
> > > > > > > > > > | SPIDER DEV    | **0.98**                      | **0.98**                   | **0.97**                         | **0.94**                        | **1.00**                   |

---

> > > > > > > > > > > ### Comment · Reviewer_U5Jd · 2023-08-29
> > > > > > > > > > >
> > > > > > > > > > > The argument on ChatGPT and the T5 results are helpful in addressing my concerns. I'm satisfactory with the efforts made by the authors. I would raise my rating and expect a revision for narrowing the scope to SQL-centric tasks and integrating the new materials (incl. experimental settings, results, and discussions) into the paper.
> > > > > > > > > > >
> > > > > > > > > > > To save space for presenting more results in Table 4, which I think are important for comparison with Spider, you may want to use in-text equations for Equations 1-5, like the metrics in Section 3.3.

---

### Official Review · Reviewer_GkAC · 2023-07-22
**Review of QATCH paper**

**Rating:** 7
**Confidence:** 4

**Strengths:**

The contribution of generating test cases from various tables is clever and immediately useful to the problem. The templated approach to focus on structural question complexity rather than linguistic complexity is appreciated.

I appreciate the discussion of details of failed tests near the end of Section 4 and what they say about the limitations of the model.

**Additional Feedback:**

I find the test generator portion of the paper particularly compelling and am interested to try it out on some of my related work.

**Clarity:**

Overall, the paper is quite clear. I identified several minor improvements in the sections above.

**Correctness:**

The claims made in the submission are correct to my knowledge. The evaluation methods and experiment design appear to be correctly performed, though I question the utility of some of the new metrics and definitely question the validity of averaging them all together as part of scoring.

**Documentation:**

There is sufficient detail to support reproducibility, including a github repo.

**Ethics:**

No, I identified no ethical concerns.

**Limitations:**

There appears to be no discussions of limitations specifically called out. In section 5, there is mention of augmenting the framework with more complex queries containing GROUP BY clauses and nested queries, which would be nice to call out explicitly.

**Opportunities For Improvement:**

My most substantive complaint is that I am struggling to understand the utility of the tuple cardinality and order metrics. It might be worth expanding on situations that make these useful.

The example in Figure 3 could be made more clear as I had to puzzle over the numbers for some time to ascertain correctness. Showing the NL question, SQL query, and a sample of the dataset might help make the data more intuitive, even if it isn’t strictly necessary. (This could also be used to illustrate the automatic generation of testing checklists through SQL in the previous section.)

Table 2 would be better placed within section 4 for greater clarity.


**Relation To Prior Work:**

Yes, there is clear and cogent discussion of prior work and where the authors derived inspiration.

**Summary And Contributions:**

Table Representation Learning models trained on large open-domain datasets can have performance characteristics that are sensitive to the content domain, schema, and quality of the data to which they are applied. This creates a challenge for industry practitioners who need to select a model that best suits their enterprises’ proprietary data, as existing open datasets and benchmarks may not effectively capture the nuances of this data.

In order to effectively evaluate a TRL model, an engineer must craft test data–a time-consuming task that is unlikely to result in a rigorous evaluation without considerable effort. The paper here proposes a toolbox to streamline this task by automatically generating and evaluating test cases from unseen proprietary data.

The authors also develop metrics beyond the commonly reported “execution accuracy” to capture other aspects of retrieval.

---

> ### Author Response · Authors · 2023-08-18
>
> **General comment**: We would like to thank the reviewer for her/his insightful comments and suggestions, which allow us to improve the quality of our work.  Following the reviewer’s indications, we will better clarify the utility of the *tuple cardinality* and *tuple order* metrics and comment on the request of more detailed results and more complex queries.
>
> **Reviewer’s concern**: Clarify the utility of the *tuple cardinality* and *tuple order* metrics.\
> **Authors’ answer**: To explain the *tuple cardinality*, let us consider the following example:
> - The expected output consists of a tuple with the following values: (a1, b1, c1).
> - The produced output 1 corresponds to the following two tuples: (a1, b1, c1), (a1, b1, c1).
> - The produced output 2 has the following two tuples: (a1, b1), (c1).
>
> The *tuple constraint* is one if the expected and produced outputs have the same schema, the same cardinality and the same cell values, zero otherwise.
> So the produced “output 1” has *tuple constraint* equal to zero because its cardinality is two instead of one, whereas the produced “output 2” has *tuple constraint* equal to zero because its schema differs from the expected one. Therefore, we introduce the *tuple cardinality* metric as a softer constraint than the *tuple constraint* since it does not consider neither the schema nor the cell values but only the cardinality. However, it has to be analysed with *cell precision* and *cell recall* to be meaningful. To complete the example, the *tuple cardinality* scores are 0.5 in both cases (two tuples returned instead of one) and *cell precision*, *cell recall* equal to one.
>
> For the *tuple order* metric, we remark that this is measured only in cases where the query (question) requires the output to be sorted (line 232 of the original submission). The other metrics follow the traditional SQL set semantics and cannot capture this aspect in the evaluation.
>
> **Reviewer’s concern**: Revision of Figure 3.\
> **Authors’ answer**: We will add the suggested details (NL question, SQL query and data sample) as suggested.
>
> **Reviewer’s concern**: Wrong position of Table 2.\
> **Authors’ answer**: We will move it as suggested.
>
> **Reviewer’s concern**: Request for more complex queries.\
> **Authors’ answer**: We will declare more clearly the lack of support of constructs such as GROUP BY and mention that also preliminary experiments with more complex SQL support the main insights derived from the current SQL subset. We report an example of a more complex query template in the following Table. The example include clauses with  GROUP BY and HAVING. The results of the tests with such template also highlight the difference between proprietary and non-proprietary tables.
> In the Semantic Parsing scenario, RESDSQL shows for this template a difference of  20 absolute points when executing the generated tests with proprietary tables compared to SPIDER tables (average of the metrics of 0.60 and 0.80, respectively).
>
> |                   |                                                                                                             |
> |-------------------|-------------------------------------------------------------------------------------------------------------|
> **Category** | *having-agg-avg-GR/LS*                                                                               |
> | **Query**    | SELECT \{$c_i$\} FROM \{*T*\} GROUP BY \{$c_i$\} HAVING AVG(\{$n_j$\}) $\geq / \leq \mu_{mean_{ij}}$ |
> | **Question** | List the \{$c_i$\} which average of \{$n_j$\} is at least/at most $\mu_{mean_i}$ in table \{*T*\}.   |
>
> *Table caption*: $T$ is the target relational table. $c_i$, $n_i$ $\in$ $S_T$ ($1 \leq i \leq n$) are respectively a categorical and a numerical attribute of the $T$'s schema. $\mu_{mean_{ij}}$ is the mean of the query result SELECT \{$c_i$\}, AVG(\{$n_j$\}) FROM \{*T*\} GROUP BY \{$c_i$\}.
>
> **Reviewer’s concern**: Request for more detailed scores.\
> **Authors’ answer**: We will revise the presentation of the main body to give more examples of the different scores (not averaged) and will report the detailed results per metric for all experiments in the supplementary material. Detailed results with the highest granularity for each category of SQL, as we have done in Table 9 (Appendix C), give the most insights about the results. However, this granularity is not needed to deliver the main results of the paper, as those are already visible in the aggregated form. We will better clarify that the averages are calculated not as (score for each SQL category) / |SQL categories|, but by averaging all metrics, i.e., the score reflects the ability of the model to answer a generic question.

---

### Decision · Program_Chairs · 2023-09-22

**Decision:**

Accept (Poster)

**Comment:**

This paper describes a framework for evaluating table representation learning models on unseen data for the tasks of question answering and semantic parsing.  It does so by generating queries against the unseen (proprietary data) and then using a set of newly introduced metrics. The test generator was highlighted as an important and useful contribution to the paper. Overall, I think this question of how to evaluate on unseen data and providing the framework to do so would be helpful for furthering the exploration of figuring out how to learn good table representations. In addition, for a benchmark and datasets I find the quality of the resource itself to be important. Here I would commend the repository itself for being clear and also guiding on how to use the framework.

I would commend the reviewers and the authors for a very informative back and forth.

I'm recommending acceptance under the assumption that the authors will make all the changes that they identified based on the reviews comments. In particular, I would call out:
- Make sure the title and abstract correctly convey the contribution (e.g. Benchmarking SQL-centric tasks with Table Representation Learning Models on Your Data)
- Add explications about the difficulty of the tasks and also the additional supporting evidence thereof.
- Add the additional clarifications about the metric definitions.